# GBA: A Tuning-free Approach to Switch between Synchronous and Asynchronous Training for Recommendation Models

**Wenbo Su**[*], **Yuanxing Zhang**,[*] **Yufeng Cai, Kaixu Ren, Pengjie Wang, Huimin Yi,**
**Yue Song, Jing Chen**[1]**, Hongbo Deng, Jian Xu, Lin Qu**[1]**, Bo Zheng**[†]
Alibaba Group
{vincent.swb, yuanxing.zyx, baike.cyf, kaixu.rkx,
pengjie.wpj, huimin.yhm, yue.song, dhb167148,
xiyu.xj, bozheng}@alibaba-inc.com
[1]{gongcheng.cj, xide.ql}@taobao.com

## Abstract

High-concurrency asynchronous training upon parameter server (PS) architecture and high-performance synchronous training upon all-reduce (AR) architecture are the most commonly deployed distributed training modes for recommendation models. Although synchronous AR training is designed to have higher training efficiency, asynchronous PS training would be a better choice for training speed when there are stragglers (slow workers) in the shared cluster, especially under limited computing resources. An ideal way to take full advantage of these two training modes is to switch between them upon the cluster status. However, switching training modes often requires tuning hyper-parameters, which is extremely time- and resource-consuming. We find two obstacles to a tuning-free approach: the different distribution of the gradient values and the stale gradients from the stragglers. This paper proposes Global Batch gradients Aggregation (GBA) over PS, which aggregates and applies gradients with the same global batch size as the synchronous training. A token-control process is implemented to assemble the gradients and decay the gradients with severe staleness. We provide the convergence analysis to reveal that GBA has comparable convergence properties with the synchronous training, and demonstrate the robustness of GBA the recommendation models against the gradient staleness. Experiments on three industrial-scale recommendation tasks show that GBA is an effective tuning-free approach for switching. Compared to the state-of-the-art derived asynchronous training, GBA achieves up to 0.2% improvement on the AUC metric, which is significant for the recommendation models. Meanwhile, under the strained hardware resource, GBA speeds up at least 2.4x compared to synchronous training.

## 1 Introduction

Nowadays, recommendation models with a large volume of parameters and high computational complexity have become the mainstream in the deep learning communities [13]. Accelerating the training of these recommendation models is a trending issue, and recently synchronous training upon high-performance computing (HPC) has dominated the training speed records [18, 16, 31]. The resource requirements of the synchronous training upon AR are more rigorous than the asynchronous

---

[*]∗ These authors contributed equally to this work.

[†]† Corresponding author

36th Conference on Neural Information Processing Systems (NeurIPS 2022).

training upon PS [1]. In a shared training cluster with dynamic status [3], the synchronous training would be retarded by a few straggling workers. Thus, its training speed may be even much slower than the high-concurrency asynchronous training.

Should it be possible to switch the training mode according to the cluster status, we will have access to making full use of the limited hardware resources. Switching the training mode for a specific model usually demands tuning of the hyper-parameters for guarantees of accuracy. Re-tuning the hyper-parameters is common in the one-shot training workloads (e.g., the general CV or NLP workloads) [20]. However, it is not applicable for the continual learning or the lifelong training of the recommendation models [10], as tuning would be highly time- and resource-consuming. When switching the training mode of representative recommendation models, we confront three main challenges from our shared cluster: 1) Model accuracy may suffer from a sudden drop after switching, requiring the model to be retrained on a large amount of data to reach the comparable accuracy before switching; 2) The distribution of gradient values is different between synchronous training and asynchronous training, making the models under two training modes difficult to reach the same accuracy by tuning the hyper-parameters;3) The cluster status imposes staleness on the asynchronous training, and staleness negatively impacts the aggregation of gradients, especially for the dense parameters.

We conduct a systematic investigation of the training workloads of recommendation models to tackle the above challenges. It is found that when the global batch size (i.e., the actual batch size of gradient aggregation) is the same, the distribution of gradient values of asynchronous training tends to be similar to that of synchronous training. Besides, we notice that due to the high sparsity, the embedding parameters in recommendation models are less frequently updated than the dense parameters, leading to a stronger tolerance for staleness than the general CV or NLP deep learning models. Based on these insights, we propose Global Batch gradients Aggregation (GBA), which ensures the model keeps the same global batch size when switched between the synchronous and asynchronous training. GBA is implemented by a token-control mechanism, which resorts to bounding the staleness and making gradient aggregation [12]. The mechanism suppresses the staleness following a staleness decay strategy over the token index. The faster nodes would take more tokens without waiting, and thereby GBA trains as fast as the asynchronous mode. Furthermore, the convergence analysis shows that GBA has comparable convergence properties with the synchronous mode, even under high staleness for recommendation models. We conduct an extensive evaluation on three continual learning of recommendation tasks. The results reveal that GBA performs well on both accuracy and efficiency with the same hyper-parameters. Particularly, GBA improves the AUC metric by 0.2% on average compared to the state-of-the-art training modes of asynchronous training. Besides, GBA presents at least 2.4x speedup over the synchronous AR training in the cluster with strained hardware resources.

To the best of our knowledge, this is the first work to approach switching between synchronous and asynchronous training without tuning the hyper-parameters. GBA has been deployed in our shared training cluster. The tuning-free switching enables our users to dynamically change the training modes between GBA and the synchronous HPC training for the continual learning tasks. The overall training efficiency of these training workloads is thereby significantly improved, and the hardware utilization within the cluster is also raised by a large margin.

## 2   Related Work

**Distributed training mode**. PS [21] and AR [18] are two mainstream architectures for the training workloads of recommendation models, accompanied by the asynchronous training and synchronous training, respectively. Researchers are enthusiastic about the pipeline, communication, and computation optimization for the AR architecture of recommender systems [31]. Meanwhile, to improve the training efficiency of the PS architecture, researchers propose a category of *semi-synchronous* training mode [12]. For example, Hop-BS [24] restricts the gradient updates under the bounded staleness, and Hop-BW [24] ignores the gradients from the stragglers with the well-shuffled and redundancy data. Recently, a category of *decentralized training* has been proposed in many studies to scale out the AR architecture. Local all-reduce [25], local update [26], exponential graph [28], and many topology-aware solutions have proven promising in the NLP and CV tasks. However, owing to the sparsity in recommendation models, the inconsistent parameters among workers and the dropped gradients of the scarce IDs would intolerably degrade the accuracy. Besides, these training modes

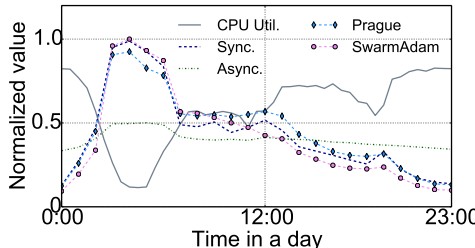

Figure 1: Normalized QPS of four training modes in training YouTubeDNN models in a shared cluster, with CPU utilization in a day.

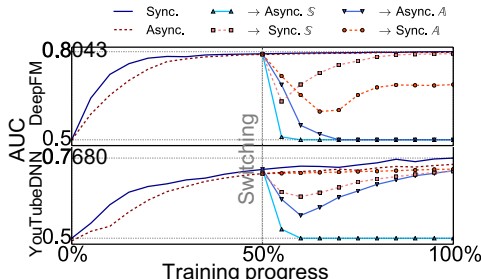

Figure 2: The AUC on the validation set of Criteo-4GB and Private by every 5% progress during training, switching at 50% progress.

hardly consider the requirements to switch to another training mode according to the cluster status, though switching is beneficial to improve the training efficiency in the shared training clusters.

**Staleness and noisy gradients**. The indeterminate or even inferior model accuracy of asynchronous training is mainly attributed to the staleness [7] and the noisy gradients [27] caused by the small batch. Although prior research has pointed out that the converged giant model is less sensitive to staleness [8], staleness is still a negative factor in the accuracy of the continual recommendation training. Many efforts have been put into controlling staleness via Taylor expansion [32], weighted penalty [35], etc. Recent works present a large-batch training paradigm with specially-designed optimizers to scale the gradients before updating [29], and point out that it can reach the best accuracy by merely adjusting batch size [11, 30]. There are also attempts to change gradient aggregation strategies during the asynchronous training to achieve stable model accuracy [22]. GBA generalizes the staleness control paradigm to the recommendation workloads by token-control mechanism, which finds the balance between bounding staleness and ignoring gradients. GBA runs with the same global batch size as the synchronous mode, ensuring the effective switching between GBA and synchronous training without tuning hyper-parameters.

## 3 Preliminaries

### 3.1 Distributed Training of Recommendation Models

Recommendation models usually comprise two modules: the *sparse* module contains the embedding layers with the *embedding parameters*, mapping the categorical IDs into numerical space; the *dense* module contains the computational blocks with the *dense parameters*, such as attention and MLP [4, 33], to exploit the feature interactions. The main difference between the two kinds of parameters is the occurrence ratio in each training batch. Each training batch needs all the dense parameters, yet only a tiny amount of embedding parameters are required according to the feature IDs in the data shard. The latest development of recommendation models introduces high complexity and a large volume of parameters, making distributed training essential to improve training efficiency. The synchronous HPC training mode usually adopts the AR architecture, where the dense parameters are replicated, and the embedding parameters are partitioned on each worker. HPC should be deployed by monopolizing a few high-performance workers and making full use of the associated resources, which may be retarded by the slow workers [19]. PS architecture is usually coupled with asynchronous high concurrency training where the parameters are placed on PSs, and the workers are responsible for the computation. On the one hand, the high concurrency mechanism activates the fragmentary resources in the training cluster by deploying hundreds of workers. On the other hand, the asynchronous training brings in *gradient staleness*, which occurs when the gradient is calculated based on the parameters of an old version and applied to the parameters of a new version.

### 3.2 Observations and Insights within a Shared Training Cluster

We investigate the training workloads of recommendation models from a shared training cluster to observe the obstacles and necessities of switching training modes.

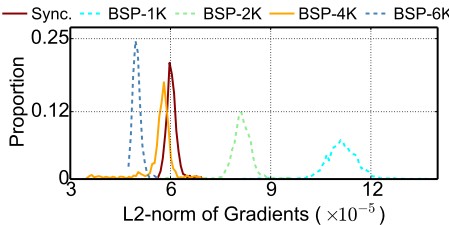
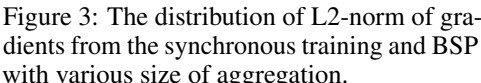

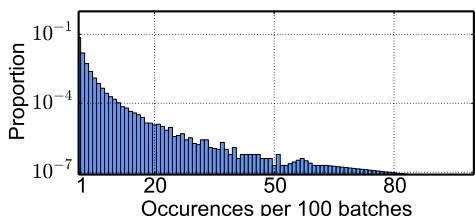

Figure 3: The distribution of L2-norm of gradients from the synchronous training and BSP with various size of aggregation.

Figure 4: The skewed distribution of ID occurrences across batches, reflecting the frequency that an ID gets updated.

**Observation 1: Cluster status determines the performance of training modes.** Figure 1 illustrates the average CPU utilization within a real shared cluster, and the corresponding samples/queries per second (QPS) of a YouTubeDNN [6] model by the synchronous and asynchronous training mode. The utilization and QPS are normalized by their maximal value, respectively. When the cluster is relatively vacant, models trained in the synchronous mode can fully occupy the hardware resources, satisfying HPC conditions and presenting high efficiency. When there are plenty of heterogeneous workloads in the cluster, slow workers dominate the training speed, making the asynchronous training mode run much faster than the synchronous mode. We also implement two approaches of local all-reduce[3]. Since the status of each device in the cluster is constantly changing, the local all-reduce-based mode would not work well when confronting resource shortages.

**Observation 2: Directly switching training mode brings sudden drop on accuracy.** We run DeepFM [23] over Criteo-4GB [15] (few parameters, fast convergence) and YouTubeDNN on Private dataset (trillions of parameters, slow convergence) in the shared cluster. We tune the hyper-parameters from scratch for the best model accuracy of both asynchronous and synchronous mode, and denote the two sets of hyper-parameters as set $\mathbb{A}$ and set $\mathbb{S}$, respectively. After training in one training mode, we evaluate the tendency of the training AUC after switching to the other training mode with set $\mathbb{A}$ or set $\mathbb{S}$. Figure 2 illustrates that after switching from the synchronous mode to the asynchronous mode, the AUC encounters sudden drop and even decreases to 0.5. The AUC drop also appears in the opposite-side switching, indicating that this condition is irrelevant to whether the model had been converged. These observations imply that directly switching the training mode requires heavy effort in re-tuning the hyper-parameter. Inherently, training modes would lead to different convergence or minima owing to the difference in batch size, learning rate and many other factors, which have already received in-depth theoretical research [2, 17]. We provide theoretical analysis to explain the sudden drop in Appendix D.

We then probe into the insights from asynchronously training recommendation models.

**Insight 1: Distribution of the gradient values is related to the aggregated batch size.** We attempt to investigate the reason for observation 2 from the gradient aspect. We implement asynchronous bulk synchronous parallel (BSP) on the YouTubeDNN recommendation task, which asynchronously aggregates $K$ gradients from workers before applying the values to the parameters. Here, we set $K$ to 100, the same as the number of workers. Besides, we compare the synchronous training in 6.4K local batch size (64 workers). Figure 3 plots the distribution of the L2-norm of gradient values from the synchronous training and BSP with various local batch sizes. It is evident that the batch size determines the mean and variance of the distribution. The distribution of BSP resembles synchronous training when the aggregation size is similar (i.e., BSP-4K). The result suggests that the same aggregation size could lead to a similar distribution of gradient values. However, there is still a gap in model accuracy after equalizing the global batch size between the asynchronous training and the synchronous training, mainly induced by gradient staleness.

**Insight 2: The gradient staleness imposes different impact on the embedding parameters and the dense parameters.** Due to the skewed distribution, most IDs would merely appear in a small number of batches, as depicted in Fig. 4. It means that in the recommendation models, only a tiny portion of IDs would be involved in every single batch, and the embedding parameters are less

---

[3]SwarmAdam is a variant of SwarmSGD [26] with Adam optimizer. It is uncommon to use SwarmAdam and Prague [25] in recommendation models as they may lead to accuracy loss which is not tolerable in the business.

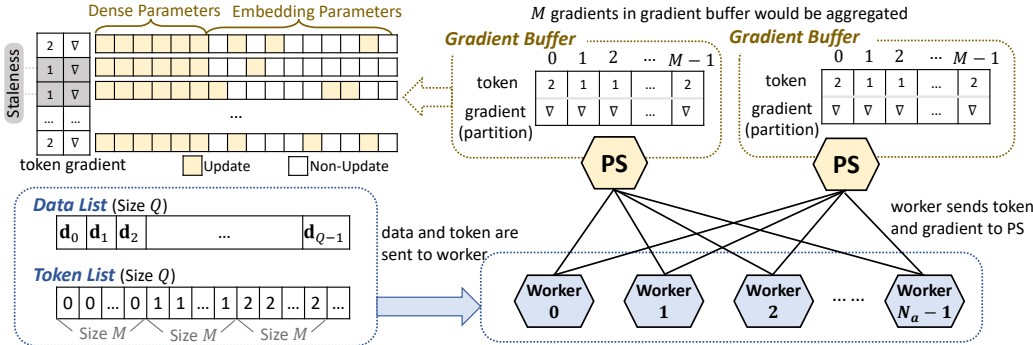

Figure 5: Illustration of the token-control mechanism in GBA: every $M$ gradients would be aggregated in the buffer before the PSs apply them to the parameters; workers report gradients to the PSs along with a token indicating the degree of data staleness.

frequently updated than the dense parameters. Therefore, the embedding parameters tend to be more robust on the gradient staleness than the dense parameters (for example, considering a worker in training, there could be five updates for the dense parameters, yet only two updates for the embedding of a specific ID).

## 4 Global Batch based Gradients Aggregation

### 4.1 Training Recommendation Models with GBA

Switching the distributed training mode for recommendation models should get rid of tuning the hyper-parameters. We introduce the concept of *global batch size*, which is defined as the actual batch size when gradients are aggregated and applied, and propose GBA for the tuning-free switching. We denote the *local batch size*, i.e., the actual batch size on each worker and the number of workers, as $B_s$ and $N_s$ for the synchronous training, $B_a$ and $N_a$ for the asynchronous training. Then, the global batch size in synchronous training, denoted by $G_s$, can be calculated as $B_s \times N_s$. Following Insight 1, GBA remains the global batch size unchanged when we switch the distributed training model from synchronous training to asynchronous training. For each step, all the dense parameters would be updated, and only a small number of embedding parameters would be updated. Then the dense parameters and embedding parameters obtain different gradient staleness during training. Hence, we define the *data staleness* as the unified staleness in training recommendation models. The data staleness describes the gap between the global step when the worker begins to ingest a data batch and the global step when the calculated gradient is applied. Obviously, the data staleness in the synchronous training mode is constantly zero. Based on data staleness, we implement GBA by a token-control mechanism on the PS architecture to cope with the sparsity and the dynamic cluster status.

Figure 5 illustrates the architecture of the proposed token-control mechanism. Over the canonical PS, we prepare a queue called *data list* to arrange the data (addresses) by batches. Given a dataset $\mathcal{D}$, suppose we can split it into $Q$ batches of size $B_a$, denoted by $\mathcal{D} = (\mathbf{d}_0, \mathbf{d}_1, \ldots, \mathbf{d}_{Q-1})$. Meanwhile, we establish another queue called *token list* to yield the token of each individual batch. The token list contains $Q$ tokens, denoted by $(t_0, t_1, \ldots, t_{Q-1})$, each attached to one batch in the data list to indicate the global step when this batch is sent to a worker. The token value starts from zero, and each token value repeats $M$ times in the token list. Here, $M$ is the number of batches we use to aggregate gradients. Under this setting, we can deduce that there will be $K = \lceil \frac{Q}{M} \rceil$ gradient updates during the training. Then we set $t_i = \lfloor \frac{i}{K} \rfloor, \forall i \in \{0, 1, \ldots, Q-1\}$ to ensure that the token list yields the token value in ascending order. Apart from the two queues, we also employ a *gradient buffer* to receive the gradients calculated by the workers with the corresponding tokens of the gradients. To be consistent with the tokens, the capacity of the gradient buffer is set to $M$, and therefore the PSs would aggregate $M$ gradients before applying them to the variables. Note that each PS maintains an individual gradient buffer to deal with the gradients of the corresponding partitions of the variables.

During the training process, a worker would pull the parameters from PS, a batch from the data list, and a token from the token list simultaneously before ingesting the data and computing the gradient locally. When a worker completes calculating the gradient of a batch, the gradient and the corresponding token are sent to the gradient buffer on PS. Then, the worker immediately proceeds to work on the next batch. In this way, the fast workers can keep working without waiting for the slow ones. When the gradient buffer reaches the capacity of $M$ pairs of gradients and tokens, all the gradients are aggregated to apply once, and at the same time, the buffer will be cleared. This is what we call finishing a global step, and thereby the global batch size in GBA can be calculated as $G_a = B_a \times M$. According to the design, we aim to keep the global batch size consistent in switching, that is, $G_s = G_a$. Hence, we can set the size of the gradient buffer to be $M = \frac{B_s \times N_s}{B_a}$. We would use $M$ workers in GBA, i.e., $N_a = M$, to avoid the intrinsic gradient staleness led by the inconsistency between the number of workers and the number of batches to aggregate.

At the update of global step $k$, denote $\tau(m, k)$ the $m$-th token in the gradient buffer. When we aggregate the gradients in the gradient buffer, we shall decay the gradients that suffer from severe staleness. GBA could employ different staleness decay strategies to mitigate the negative impact from the staleness according to the token index, and in this work we define it as:

$$f(\tau(m,k),k) = \begin{cases} 0, & k - \tau(m,k) > \iota \\ 1, & k - \tau(m,k) \leq \iota, \end{cases} \tag{1}$$

where $\iota$ is the threshold of tolerance. If $f(\tau(m,k),k) = 0$, we exclude the $m$-th gradient in the buffer due to the severe staleness; otherwise, we aggregate the gradient. As we can see, tokens help identify whether the corresponding gradients are stale and how many stale steps are behind the current global step. In this case, although the token is designed over the data staleness, the negative impact from the canonical gradient staleness can also be mitigated.

## 4.2  Convergence Analysis

We have seen much research on the convergence analysis of the synchronous and asynchronous training. Following the assumptions and convergence analysis in Dutta et al. [7], the expectation of error after $k$ steps of gradient updates in the synchronous training can be deduced by:

$$\mathbb{E}[F(\mathbf{w}_k)] - F^* \leq \frac{\eta L \sigma^2}{2cN_sB_s} + (1 - \eta c)^k (F(\mathbf{w}_0) - F^* - \frac{\eta L \sigma^2}{2cN_sB_s}), \tag{2}$$

where $\mathbf{w}_k$ denotes the parameter in step $k$, $\eta$ denotes learning rate, $L$ is the Lipschitz constant and $\sigma$ denotes the variance of gradients. $F(\mathbf{w})$ is the empirical risk function that is strongly convex with parameter $c$. $\mathbb{E}[F(\mathbf{w}_k)] - F^*$ is the expected gap of the risk function from its optimal value, used as the error after $k$ steps. As mentioned in Eqn. (2), The first term in the right, i.e. $\frac{\eta L \sigma^2}{2c(N_sB_s)}$, would be the error floor, and $(1 - \eta c)$ is the decay rate. The proposed GBA is derived upon the asynchronous gradient aggregation. We assume that, for some $\gamma \leq 1$,

$$\gamma \geq \frac{\zeta E[||\nabla F(\mathbf{w}_k) - \nabla F(\mathbf{w}_{\tau(m,k)})||_2^2]}{E[||\nabla F(\mathbf{w}_k)||_2^2]}. \tag{3}$$

Here, $\gamma$ is a measure of gradients impact induced by the staleness; smaller value of $\gamma$ indicates that staleness makes less accuracy deterioration of the model. Besides, $\zeta$ indicates the average probability that any parameter in the model would be both updated in step $k$ and step $\tau(m, k)$. Intuitively, $\zeta$ would be far below 1 in the recommendation models due to the strong sparsity. Then, the error of GBA after $k$ steps of aggregated updates would become (Appendix A presents the proof):

$$\mathbb{E}[F(\mathbf{w}_k)] - F^* \leq \frac{\eta L \sigma^2}{2c\gamma'MB_a} + (1 - \eta\gamma'c)^k (\mathbb{E}[F(\mathbf{w}_0)] - F^* - \frac{\eta L \sigma^2}{2c\gamma'MB_a}), \tag{4}$$

where $\gamma' = 1 - \gamma + \frac{p_0}{2}$ and $p_0$ is a lower bound on the conditional probability that the token equals to the global step, i.e., $\tau(m,k) = k$. Equation (4) proves the convergence of GBA. Considering the error floors of Eqn. (2) and Eqn. (4), $M \times B_a$ should be set close to $N_s \times B_s$ to make GBA tuning-free. It is exactly the global batch size we use in GBA, consistent with our main idea of keeping global batch size unchanged. Recall that with the embedding parameters, $\zeta < 1$ makes $\gamma$ lower than the training of general CV or NLP models. Consequently, the error floor remains low in GBA.

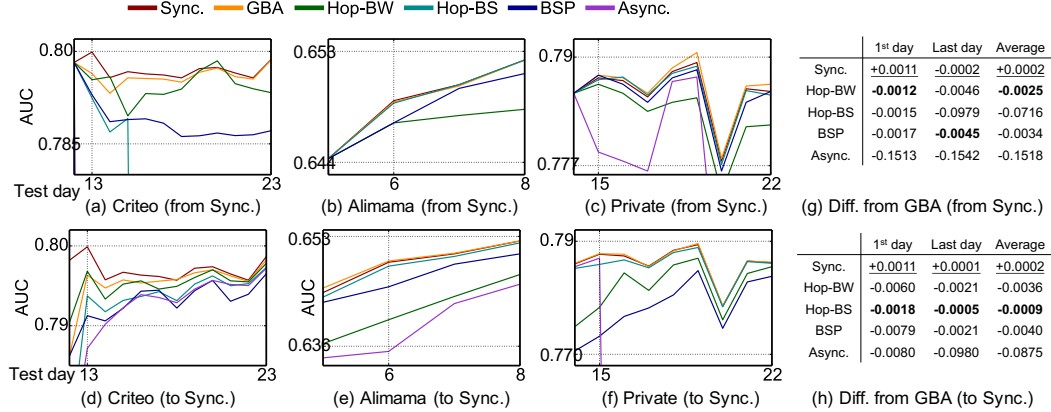

Figure 6: The AUC tendencies on the test days of the three datasets after inheriting a base model: (a-c) from the synchronous training modes and switching to the compared training modes; (d-f) from the compared training modes and switching to the synchronous training modes; (g-h) AUC difference between GBA and the other training modes after switching from/to synchronous training.

## 5 Evaluation

### 5.1 Settings

Table 1: Settings of the three continual recommendation tasks by the compared training modes.

| Task | Model description | Data parts | Sample per day | Optimizer | Learning rate | # of workers | Local batch size | Private hyper-param. |
|---|---|---|---|---|---|---|---|---|
| Criteo (DeepFM) | 19M(x40K) FLOPS 45B params. 16 avg. dim. | 12 days (base) 11 days (eval) | 190M | Adagrad (Async.) Adam (Others) | 0.006 (Async.) 0.0011 (Others) | 32 (Sync.) 100 (Others) | 5K (Async.) 12.8K (GBA) 40K (Others) | Hop-BS ($b_1$=2) BSP ($b_2$=20) Hop-BW ($b_3$=20) GBA ($\iota$=3) |
| Alimama (DIEN) | 112M(x3K) FLOPS 160B params. 19 avg. dim. | 5 days (base) 3 days (eval) | 90M | Adagrad (Async.) Adam (Others) | 0.008 (Async.) 0.0015 (Others) | 32 (Sync.) 128 (Others) | 1K (Async.) 0.75K (GBA) 3K (Others) | Hop-BS ($b_1$=2) BSP ($b_2$=20) Hop-BW ($b_3$=20) GBA ($\iota$=4) |
| Private (YouTubeDNN) | 746M(x6.4K) FLOPS 1.9T params. 24 avg. dim. | 14 days (base) 8 days (eval) | 2B | Adagrad (Async.) Adam (Others) | 0.001 (Async.) 0.0006 (Others) | 64 (Sync.) 400 (Others) | 1K (Async.) 1K (GBA) 6.4K (Others) | Hop-BS ($b_1$=2) BSP ($b_2$=50) Hop-BW ($b_3$=100) GBA ($\iota$=4) |

We conduct systematical evaluations to examine the performance of GBA and make a fine-grained analysis. The evaluations involve three industrial-scale recommendation tasks: 1) On the Criteo-1TB dataset [14], we implement DeepFM, where the hyper-parameters on Criteo-4GB (AUC 0.8043) are utilized; 2) On the Alimama dataset [9], we implement DIEN [34] and use the recommended hyper-parameters in the original design; 3) On the Private dataset, we implement YouTubeDNN, and we tune the best hyper-parameters. The models are implemented in DeepRec [5] with the expandable HashTables. Detailed information on the dataset and the models are listed in Tab. 1. We imitate the continual training without changing the hyper-parameters to the models, and ensure a similar cluster status for all evaluations. Inheriting from a pre-trained checkpoint, we repeatedly train on the data of every day and evaluate the data of the subsequent day. The training cluster is equipped with a Tesla-V100S GPU and Skylake CPU. We focus on AUC as the accuracy metric and global/local QPS (QPS of all/single workers) as the efficiency metric.

We select several state-of-the-art PS-based training modes: Bounded staleness (Hop-BS) restricts the maximal differences of gradient version between the fastest and the slowest workers, controlled by $b_1$; Bulk synchronous parallel (BSP) aggregates a pre-set number $b_2$ of gradients when applying gradients to the parameters, regardless of the gradient version; Backup worker (Hop-BW) ignores the pre-set number $b_3$ of gradients from the slowest workers during each gradient aggregation. We enumerate the specialized hyper-parameters of each training mode and record the statistics when reaching its best AUC.

Table 2: Global QPS of the compared training mode on the three tasks.

|  | Sync. | Async. | Hop-BS | BSP | Hop-BW | GBA |
|---|---|---|---|---|---|---|
| **Criteo** | 1,436K($\pm$224K) | 3,253K($\pm$84K) | 2,227K($\pm$336K) | 3,247K($\pm$93K) | 2,559K($\pm$294K) | **3,240K($\pm$97K)** |
| **Alimama** | 182K($\pm$52K) | 403K($\pm$33K) | 217K($\pm$65K) | 403K($\pm$33K) | 288K($\pm$48K) | **399K($\pm$35K)** |
| **Private** | 43K($\pm$21K) | 90K($\pm$15K) | 29K($\pm$11K) | 88K($\pm$17K) | 66K($\pm$24K) | **87K($\pm$19K)** |

Table 3: Fine-grained analysis between GBA and the other training modes.

| Local QPS | | AUC | | # of drop | | Avg. grad. staleness (max) | | |
|---|---|---|---|---|---|---|---|---|
| Async. | GBA | Sync. | GBA | Hop-BW | GBA | Hop-BS | GBA | BSP |
| 78K($\pm$23K) | 74K($\pm$25K) | 0.7864 | 0.7864 | 300K | 1,454 | 0.06 (2) | 0.21 (11) | 2.61 (12) |
| 90K($\pm$15K) | 87K($\pm$19K) | 0.7864 | 0.7866 | 300K | 898 | 0.04 (2) | 0.15 (11) | 1.92 (12) |
| 99K($\pm$12K) | 98K($\pm$12K) | 0.7864 | 0.7865 | 300K | 786 | 0.03 (2) | 0.12 (9) | 1.62 (10) |

## 5.2 Performance of Training Modes

We first examine the performance of GBA. Figure 6(a-c) records the AUC tendencies after switching from synchronous to the other training modes over the three recommendation tasks. We mainly focus on the AUC at the first day and the last day, as well as the average AUC scores throughout the datasets. Although Hop-BW eliminates staleness, the ignorance of a large volume of data makes it perform the worst (also taken as evidence why we tend not to use local all-reduce in training recommender systems). The manipulation of global batch size contributes to the best performance on both sides of switching. Compared to the best baselines (i.e., Hop-BW), GBA improves AUC by at least 0.2% on average over the three datasets, which has the potential to increase by 1% revenue in the real-world business. Meanwhile, after switching from synchronous training, GBA obtains immediate good accuracy (AUC at the first day), while there are explicit re-convergence on the other training modes, as depicted in Figure 6(g).

Figure 6(d-f,h) illustrates the AUCs of these training modes after switching to synchronous training. We can see that GBA tends to obtain at least equal accuracy to the continuous synchronous training without switching. On the contrary, the models inherited from the other baselines require consuming more data to reach the desired accuracy of the synchronous training. The tendency of the AUC gaps between the synchronous training and the compared training modes reflects that the parameters trained by GBA are the most compatible with the synchronous training. It verifies that switching from GBA to synchronous training is also tuning-free.

We collect metrics of the training efficiency during the above experiments, and report their global QPS in Tab. 2. The results reveal that GBA performs similarly to the asynchronous training. Although Hop-BS works better than BSP and Hop-BW in accuracy, it struggles to deal with the slow workers. It indicates that when facing a resource shortage in the shared cluster, GBA can provide similar accuracy with synchronous training mode, while running as fast as the asynchronous training mode.

## 5.3 Fine-grained Analysis

We further probe into the performance of GBA. Here, we take the recommendation task on the Private dataset (the most complex model) as an example, switching from the synchronous mode to GBA.

We first analyze how the cluster status affects the performance of GBA. The experiments are repeated in the cluster during different periods of a day. We collect AUC, QPS, average gradient staleness on the dense parameters (for fair comparison among the baselines), and the number of excluded batches, as shown in Tab. 3. From the results, we can infer that GBA properly finds the balance between the staleness and the excluded data (as defined in Eqn. (1)), i.e., excluding fewer data compared to Hop-BW and suppressing the staleness to the same level of Hop-BS. GBA also shows strong robustness on the dynamic cluster status, obtaining stable performance on AUC.

We then examine the impact of the batch sizes in GBA. Figure 7 depicts the average AUC score and the global QPS when we modify the local batch size (the number of workers is thereby changed) and keep the global batch size unchanged. Considering the hardware limitation on worker and the communication overhead on PS, we vary the number of workers from 100 to 800. We can see a steady state of the AUC score (i.e., absolute difference less than $10^{-4}$), while the training achieves

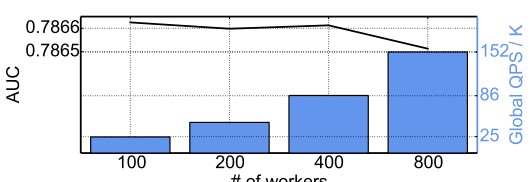
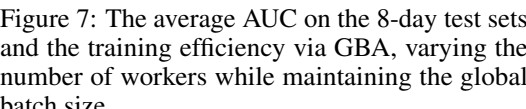
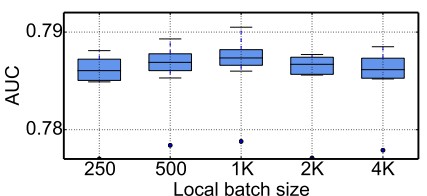

Figure 7: The average AUC on the 8-day test sets and the training efficiency via GBA, varying the number of workers while maintaining the global batch size.

Figure 8: The range of AUC on the 8-day test sets via GBA of 400 workers, varying the local batch size.

a significant efficiency boost when using more workers. It can thereby be inferred that GBA has a good capability of scaling out. Besides, we fix the number of workers to 400 and change the local batch size for each worker, which means that the global batch size would differ. As shown in Fig. 8, the inconsistent global batch size with the synchronous training makes the training encounter lower AUC scores after switching. Although the larger global batch size may have the potential to achieve better accuracy (i.e., owing to the stable and accurate gradient), the experiment indicates the model would hardly reach its best accuracy without tuning. These results verify that using the same global batch size in GBA as in the synchronous training is necessary to get rid of tuning when switching the training mode.

## 6   Conclusion

A tuning-free switching approach is demanded to take full advantage of the synchronous and asynchronous training, which can improve the training efficiency in the shared cluster. We raise insights from the investigation over the production training workloads that the inconsistent batch size and the gradient staleness are two main reasons to fail the switching regarding the model accuracy. Then GBA training mode is proposed for asynchronously training recommendation models via aggregating gradients by the global batch size. GBA enables switching between synchronous training and asynchronous training of the continual learning tasks with the accuracy and efficiency guarantees. With GBA, users can freely switch the training modes according to the status of the shared training clusters, without tuning hyper-parameters. GBA is implemented through a token-control mechanism to ensure that the faster worker should contribute more gradients to the aggregation while the gradients from the straggling workers would be decayed. Evaluations of three representative continual training tasks of recommender systems reveal that GBA achieves similar accuracy with the synchronous training, while resembling the efficiency of the canonical asynchronous training. Currently, GBA requires the users to select the training mode according to their own judgment on the cluster status. In the future, we will attempt to make GBA be adaptive to the cluster status. The guidelines of automatic switching would be derived from more analyses upon the training trace logs. It could be formulated as an optimization problem under many control factors including but not limited to the overall QPS, training cost, and task scheduling with priority.

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
