# Appendix

## A    Proof of Equation (4)

**Assumption.** Throughout the paper, we make the following assumptions:

1. $F(w)$ is an L-smooth function, i.e., $F(w_1) \leq F(w_2) + (w_1 - w_2)^T \nabla F(w_2) + \frac{L}{2}\|w_1 - w_2\|_2^2$;
2. $F(w)$ is strongly convex with parameter $c$, i.e., $2c(F(w) - F^*) \leq \|\nabla F(w)\|_2^2$;
3. The stochastic gradient $g(w_{\tau(m,k)})$ is an unbiased estimate of the true gradient, i.e.,

$$E(g(w_{\tau(m,k)})) = E(\nabla F(w_{\tau(m,k)}));$$

4. The variance of the stochastic gradient $g(w_{\tau(m,k)})$ in the asynchronous training is bounded as

$$E\left(\|g(w_{\tau(m,k)}) - \nabla F(w_{\tau(m,k)})\|_2^2\right) \leq \frac{\sigma^2}{B_a} + \frac{\Theta}{B_a}E(\|\nabla F(w_{\tau(m,k)})\|_2^2),$$

and in the synchronous training is bounded as

$$E\left(\|g(w_{\tau(m,k)}) - \nabla F(w_{\tau(m,k)})\|_2^2\right) \leq \frac{\sigma^2}{B_s} + \frac{\Theta}{B_s}E(\|\nabla F(w_{\tau(m,k)})\|_2^2).$$

**Theorem 1.** *Based on the above Assumption and $\eta \leq \frac{1}{2L(\frac{\Theta}{MB_a}+1)}$, also suppose that for some $\gamma \leq 1$,*

$$E\left(\|\nabla F(w_k) - \nabla F(w_{\tau(m,k)})\|_2^2\right) \leq \gamma E(\|\nabla F(w_k)\|_2^2),$$

*thus the expectation of error after $k+1$ steps of gradient updates **in the asynchronous training** is deduced by*

$$E(F(w_{k+1}) - F^*) \leq \frac{\eta L\sigma^2}{2c\gamma' MB_a} + (1 - \eta\gamma'c)^{k+1}\left(E(F(w_0) - F^*) - \frac{\eta L\sigma^2}{2c\gamma' MB_a}\right),$$

*where $\gamma' = 1 - \gamma + \frac{p_0}{2}$, $p_0$ is a lower bound on the conditional probability that the token equals to the global step, i.e., $\tau(m, k) = k$.*

**Corollary 1.** *To characterize the strong sparsity in the recommendation models, we suppose that for some $\gamma \leq 1$,*

$$\zeta E(\|\nabla F(w_k) - \nabla F(w_{\tau(m,k)})\|_2^2) \leq \gamma E(\|\nabla F(w_k)\|_2^2),$$

$$\gamma = \begin{cases} \gamma, & \varsigma = 1, \\ \zeta\gamma, & \varsigma \neq 1, \end{cases} \tag{1}$$

*thus the expectation of error after $k+1$ steps of gradient updates **in the asynchronous training** is*

$$E(F(w_{k+1}) - F^*) \leq \frac{\eta L\sigma^2}{2c\rho MB_a} + (1 - \eta\rho c)^{k+1}\left(E(F(w_0) - F^*) - \frac{\eta L\sigma^2}{2c\rho MB_a}\right),$$

*where $\rho = 1 - p_1\gamma - (1 - p_1)\zeta\gamma + \frac{p_0}{2}$, $\varsigma = \mathbb{I}_{dense}(parameter)$, $\mathbb{I}_A(x)$ is the indicator function of whether $x$ belongs to $A$, $p_1 = P(\varsigma = 1)$.*

It is worth noting that we have $\rho > 0$ and $\rho > \gamma'$ owing to $\zeta\gamma < \gamma \leq 1$. Since $\frac{2+p_0-2\zeta\gamma}{2(\gamma-\zeta\gamma)} > 1$, $P(\varsigma = 1) < \frac{2+p_0-2\zeta\gamma}{2(\gamma-\zeta\gamma)}$. The model GBA achieves a better performance in terms of dense and sparse parameters, like the error floor $\frac{\eta L\sigma^2}{2c\rho MB_a}$ is smaller than $\frac{\eta L\sigma^2}{2c\gamma' MB_a}$ mentioned in Theorem 1, and the convergence speed is more quickly. Clearly, the value of $p_1$ is differ for various distributions and accordingly the values of $\frac{\eta L\sigma^2}{2c\rho MB_a}$ and $1 - \eta\rho c$ are different.

**Theorem 2.** *Based on the above Assumption and $\eta \leq \frac{1}{2L(\frac{\Theta}{N_s B_s}+1)}$, the expectation of error after $k+1$ steps of gradient updates **in the synchronous training** is deduced by*

$$E(F(w_{k+1}) - F^*) \leq \frac{\eta L\sigma^2}{2cN_s B_s} + (1 - \eta c)^{k+1}\left(E(F(w_0) - F^*) - \frac{\eta L\sigma^2}{2cN_s B_s}\right).$$

To provide the proofs of Theorem 1, Corollary 1, and Theorem 2, we first prove the following lemmas.

**Lemma 1.** *Let $g(w_{\tau(i,k)})$ denote the i-th gradient of k-th global step, and assume its expectation $E(g(w_{\tau(i,k)})) = E(\nabla F(w_{\tau(i,k)}))$. Then*

$$E\Big(\|g(w_{\tau(i,k)}) - \nabla F(w_k)\|_2^2\Big)$$

$$= E(\|g(w_{\tau(i,k)})\|_2^2) - E(\|\nabla F(w_{\tau(i,k)})\|_2^2) + E\Big(\|\nabla F(w_{\tau(i,k)}) - \nabla F(w_k)\|_2^2\Big).$$

*proof of Lemma 1.*

$$E\Big(\|g(w_{\tau(i,k)}) - \nabla F(w_k)\|_2^2\Big)$$

$$= E\Big(\|g(w_{\tau(i,k)}) - \nabla F(w_{\tau(i,k)}) + \nabla F(w_{\tau(i,k)}) - \nabla F(w_k)\|_2^2\Big)$$

$$= E\Big(\|g(w_{\tau(i,k)}) - \nabla F(w_{\tau(i,k)})\|_2^2\Big) + E\Big(\|\nabla F(w_{\tau(i,k)}) - \nabla F(w_k)\|_2^2\Big) \tag{2}$$

$$+ 2E\Big(\Big\langle g(w_{\tau(i,k)}) - \nabla F(w_{\tau(i,k)}), \nabla F(w_{\tau(i,k)}) - \nabla F(w_k)\Big\rangle\Big).$$

Since $E(g(w_{\tau(i,k)})) = E(\nabla F(w_{\tau(i,k)}))$,

$$E\Big(\Big\langle g(w_{\tau(i,k)}) - \nabla F(w_{\tau(i,k)}), \nabla F(w_{\tau(i,k)}) - \nabla F(w_k)\Big\rangle\Big) = 0.$$

Returning to (2), we have

$$E\Big(\|g(w_{\tau(i,k)}) - \nabla F(w_k)\|_2^2\Big)$$

$$= E\Big(\|g(w_{\tau(i,k)}) - \nabla F(w_{\tau(i,k)})\|_2^2\Big) + E\Big(\|\nabla F(w_{\tau(i,k)}) - \nabla F(w_k)\|_2^2\Big)$$

$$= E(\|g(w_{\tau(i,k)})\|_2^2) + E(\|\nabla F(w_{\tau(i,k)})\|_2^2) - 2E\Big(\langle g(w_{\tau(i,k)}), \nabla F(w_{\tau(i,k)})\rangle\Big) \tag{3}$$

$$+ E\Big(\|\nabla F(w_{\tau(i,k)}) - \nabla F(w_k)\|_2^2\Big)$$

$$= E(\|g(w_{\tau(i,k)})\|_2^2) - E(\|\nabla F(w_{\tau(i,k)})\|_2^2) + E\Big(\|\nabla F(w_{\tau(i,k)}) - \nabla F(w_k)\|_2^2\Big).$$

$$\square$$

**Lemma 2.** *Let $v_k = \frac{1}{M}\sum_{i=1}^{M} g(w_{\tau(i,k)})$, and suppose the variance of $g(w_{\tau(i,k)})$ is bounded as*

$$E\Big(\|g(w_{\tau(i,k)}) - \nabla F(w_{\tau(i,k)})\|_2^2\Big) \leq \frac{\sigma^2}{B_a} + \frac{\Theta}{B_a}\|\nabla F(w_{\tau(i,k)})\|_2^2.$$

*Then the sum of $g(w_{\tau(i,k)})$ is bounded as follows*

$$E(\|v_k\|_2^2) \leq \frac{\sigma^2}{MB_a} + \sum_{i=1}^{M}\frac{\Theta}{M^2 B_a}E(\|\nabla F(w_{\tau(i,k)})\|_2^2) + \frac{1}{M}\sum_{i=1}^{M}E(\|\nabla F(w_{\tau(i,k)})\|_2^2).$$

*proof of Lemma 2.*

$$E(\|v_k\|_2^2) = E\left(\|\frac{1}{M}\sum_{i=1}^{M}g(w_{\tau(i,k)})\|_2^2\right) = \frac{1}{M^2}E\left(\|\sum_{i=1}^{M}g(w_{\tau(i,k)})\|_2^2\right)$$

$$= \frac{1}{M^2}E\left(\|\sum_{i=1}^{M}(g(w_{\tau(i,k)}) - \nabla F(w_{\tau(i,k)})) + \sum_{i=1}^{M}\nabla F(w_{\tau(i,k)})\|_2^2\right)$$

$$= \frac{1}{M^2}E\left(\|\sum_{i=1}^{M}(g(w_{\tau(i,k)}) - \nabla F(w_{\tau(i,k)}))\|_2^2\right) + \frac{1}{M^2}E\left(\|\sum_{i=1}^{M}\nabla F(w_{\tau(i,k)})\|_2^2\right) \tag{4}$$

$$+ \frac{2}{M^2}E\left(\Big\langle\sum_{i=1}^{M}(g(w_{\tau(i,k)}) - \nabla F(w_{\tau(i,k)})), \sum_{i=1}^{M}\nabla F(w_{\tau(i,k)})\|_2^2\Big\rangle\right)$$

Owing to $E(g(w_{\tau(i,k)})) = E(\nabla F(w_{\tau(i,k)}))$,

$$
\begin{aligned}
E(\|v_k\|_2^2) &= \frac{1}{M^2}E\left(\|\sum_{i=1}^{M}(g(w_{\tau(i,k)}) - \nabla F(w_{\tau(i,k)}))\|_2^2\right) + \frac{1}{M^2}E\left(\|\sum_{i=1}^{M}\nabla F(w_{\tau(i,k)})\|_2^2\right) \\
&= \frac{1}{M^2}\sum_{i=1}^{M}E\left(\|g(w_{\tau(i,k)}) - \nabla F(w_{\tau(i,k)})\|_2^2\right) + \frac{1}{M^2}E\left(\|\sum_{i=1}^{M}\nabla F(w_{\tau(i,k)})\|_2^2\right) \\
&\quad + \frac{2}{M^2}\sum_{i=1}^{M-1}\sum_{j=i+1}^{M}E\left(\left\langle g(w_{\tau(i,k)}) - \nabla F(w_{\tau(i,k)}), g(w_{\tau(j,k)}) - \nabla F(w_{\tau(j,k)})\right\rangle\right) \quad (5) \\
&= \frac{1}{M^2}\sum_{i=1}^{M}E\left(\|g(w_{\tau(i,k)}) - \nabla F(w_{\tau(i,k)})\|_2^2\right) + \frac{1}{M^2}E\left(\|\sum_{i=1}^{M}\nabla F(w_{\tau(i,k)})\|_2^2\right) \\
&\leq \frac{\sigma^2}{MB_a} + \sum_{i=1}^{M}\frac{\Theta}{M^2 B_a}E(\|\nabla F(w_{\tau(i,k)})\|_2^2) + \frac{1}{M^2}E\left(\|\sum_{i=1}^{M}\nabla F(w_{\tau(i,k)})\|_2^2\right).
\end{aligned}
$$

The second term in (5) could be obtained by

$$
\begin{aligned}
&E\left(\|\sum_{i=1}^{M}\nabla F(w_{\tau(i,k)})\|_2^2\right) \\
&= \sum_{i=1}^{M}E(\|\nabla F(w_{\tau(i,k)})\|_2^2) + \sum_{i=1}^{M-1}\sum_{j=i+1}^{M}2E(\langle\nabla F(w_{\tau(i,k)}), \nabla F(w_{\tau(j,k)})\rangle) \\
&\overset{(a)}{\leq} \sum_{i=1}^{M}E(\|\nabla F(w_{\tau(i,k)})\|_2^2) + \sum_{i=1}^{M-1}\sum_{j=i+1}^{M}E\left(\|\nabla F(w_{\tau(i,k)})\|_2^2 + \|\nabla F(w_{\tau(j,k)})\|_2^2\right) \quad (6) \\
&= \sum_{i=1}^{M}E(\|\nabla F(w_{\tau(i,k)})\|_2^2) + \sum_{i=1}^{M}(M-1)E(\|\nabla F(w_{\tau(i,k)})\|_2^2) \\
&= \sum_{i=1}^{M}ME(\|\nabla F(w_{\tau(i,k)})\|_2^2).
\end{aligned}
$$

Here step $(a)$ follows from $2\langle x, y\rangle \leq \|x\|_2^2 + \|y\|_2^2$.

Based on (5) and (6), we have

$$
E(\|v_k\|_2^2) \leq \frac{\sigma^2}{MB_a} + \sum_{i=1}^{M}\frac{\Theta}{M^2 B_a}E(\|\nabla F(w_{\tau(i,k)})\|_2^2) + \frac{1}{M}\sum_{i=1}^{M}E(\|\nabla F(w_{\tau(i,k)})\|_2^2).
$$

$\square$

**Lemma 3.** *Suppose $p_0$ is a lower bound on the conditional probability that the token equals to the global step, i.e., $\tau(i, k) = k$, thus*

$$
E(\|\nabla F(w_{\tau(i,k)})\|_2^2) \geq p_0 E(\|\nabla F(w_k)\|_2^2).
$$

*proof of Lemma 3.*

$$
\begin{aligned}
E(\|\nabla F(w_{\tau(i,k)})\|_2^2) &= p_0 E\left(\|\nabla F(w_{\tau(i,k)})\|_2^2 \mid \tau(i,k) = k\right) \\
&\quad + (1 - p_0)E\left(\|\nabla F(w_{\tau(i,k)})\|_2^2 \mid \tau(i,k) \neq k\right) \quad (7) \\
&\geq p_0 E(\|\nabla F(w_k)\|_2^2).
\end{aligned}
$$

$\square$

Next, we will provide the proofs of Theorem 1, Corollary 1, and Theorem 2.

*proof of Theorem 1.* Let $w_{k+1} = w_k - \eta v_k$, $v_k = \frac{1}{M} \sum_{i=1}^{M} g(w_{\tau(i,k)})$, we have

$$F(w_{k+1}) \leq F(w_k) + (w_{k+1} - w_k)^T \nabla F(w_k) + \frac{L}{2} \|w_{k+1} - w_k\|_2^2$$

$$\leq F(w_k) + \langle -\eta v_k, \nabla F(w_k) \rangle + \frac{L}{2} \eta^2 \|v_k\|_2^2 \qquad (8)$$

$$= F(w_k) - \frac{\eta}{M} \sum_{i=1}^{M} \langle g(w_{\tau(i,k)}), \nabla F(w_k) \rangle + \frac{L}{2} \eta^2 \|v_k\|_2^2.$$

Owing to $2\langle x, y \rangle = \|x\|_2^2 + \|y\|_2^2 - \|x - y\|_2^2$, (8) is shown as follows,

$$F(w_{k+1}) \leq F(w_k) - \frac{\eta}{M} \sum_{i=1}^{M} \left( \frac{1}{2} \|g(w_{\tau(i,k)})\|_2^2 + \frac{1}{2} \|\nabla F(w_k)\|_2^2 - \frac{1}{2} \|g(w_{\tau(i,k)}) - \nabla F(w_k)\|_2^2 \right) + \frac{L}{2} \eta^2 \|v_k\|_2^2$$

$$(9)$$

$$= F(w_k) - \frac{\eta}{2} \|\nabla F(w_k)\|_2^2 - \frac{\eta}{2M} \sum_{i=1}^{M} \|g(w_{\tau(i,k)})\|_2^2 + \frac{\eta}{2M} \sum_{i=1}^{M} \|g(w_{\tau(i,k)}) - \nabla F(w_k)\|_2^2 + \frac{L}{2} \eta^2 \|v_k\|_2^2.$$

Taking expectation,

$$E(F(w_{k+1})) \leq E(F(w_k)) - \frac{\eta}{2} E(\|\nabla F(w_k)\|_2^2) - \frac{\eta}{2M} \sum_{i=1}^{M} E(\|g(w_{\tau(i,k)})\|_2^2)$$

$$+ \frac{\eta}{2M} \sum_{i=1}^{M} E\left( \|g(w_{\tau(i,k)}) - \nabla F(w_k)\|_2^2 \right) + \frac{L}{2} \eta^2 E(\|v_k\|_2^2)$$

$$\overset{(a)}{=} E(F(w_k)) - \frac{\eta}{2} E(\|\nabla F(w_k)\|_2^2) - \frac{\eta}{2M} \sum_{i=1}^{M} E(\|g(w_{\tau(i,k)})\|_2^2) + \frac{\eta}{2M} \sum_{i=1}^{M} E(\|g(w_{\tau(i,k)})\|_2^2)$$

$$- \frac{\eta}{2M} \sum_{i=1}^{M} E(\|\nabla F(w_{\tau(i,k)})\|_2^2) + \frac{\eta}{2M} \sum_{i=1}^{M} E\left( \|\nabla F(w_{\tau(i,k)}) - \nabla F(w_k)\|_2^2 \right) + \frac{L}{2} \eta^2 E(\|v_k\|_2^2)$$

$$\overset{(b)}{\leq} E(F(w_k)) - \frac{\eta}{2} E(\|\nabla F(w_k)\|_2^2) - \frac{\eta}{2M} \sum_{i=1}^{M} E(\|\nabla F(w_{\tau(i,k)})\|_2^2)$$

$$+ \frac{\eta}{2} \gamma E(\|\nabla F(w_k)\|_2^2) + \frac{L}{2} \eta^2 E(\|v_k\|_2^2) \qquad (10)$$

$$= E(F(w_k)) - \frac{\eta}{2} (1 - \gamma) E(\|\nabla F(w_k)\|_2^2) - \frac{\eta}{2M} \sum_{i=1}^{M} E(\|\nabla F(w_{\tau(i,k)})\|_2^2) + \frac{L}{2} \eta^2 E(\|v_k\|_2^2)$$

$$\overset{(c)}{\leq} E(F(w_k)) - \frac{\eta}{2} (1 - \gamma) E(\|\nabla F(w_k)\|_2^2) - \frac{\eta}{2M} \sum_{i=1}^{M} E(\|\nabla F(w_{\tau(i,k)})\|_2^2)$$

$$+ \frac{L}{2} \eta^2 \left( \frac{\sigma^2}{MB_a} + \sum_{i=1}^{M} \frac{\Theta}{M^2 B_a} E(\|\nabla F(w_{\tau(i,k)})\|_2^2) + \frac{1}{M} \sum_{i=1}^{M} E(\|\nabla F(w_{\tau(i,k)})\|_2^2) \right)$$

$$= E(F(w_k)) - \frac{\eta}{2} (1 - \gamma) E(\|\nabla F(w_k)\|_2^2) + \frac{L\eta^2 \sigma^2}{2MB_a}$$

$$- \frac{\eta}{2M} \sum_{i=1}^{M} \left( 1 - \frac{L\eta\Theta}{MB_a} - L\eta \right) E(\|\nabla F(w_{\tau(i,k)})\|_2^2).$$

Here step (a) follows from Lemma 1, step (b) follow from $E(\|\nabla F(w_k) - \nabla F(w_{\tau(m,k)})\|_2^2) \leq \gamma E(\|\nabla F(w_k)\|_2^2)$, and step (c) follows from Lemma 2.

Since $\eta \leq \frac{1}{2L(\frac{\Theta}{MB_a}+1)}$, (10) could be obtained by

$$E(F(w_{k+1})) \leq E(F(w_k)) - \frac{\eta}{2}(1-\gamma)E(\|\nabla F(w_k)\|_2^2) + \frac{L\eta^2\sigma^2}{2MB_a} - \frac{\eta}{4M}\sum_{i=1}^{M}E(\|\nabla F(w_{\tau(i,k)})\|_2^2)$$

$$\overset{(d)}{\leq} E(F(w_k)) - \frac{\eta}{2}(1-\gamma)E(\|\nabla F(w_k)\|_2^2) + \frac{L\eta^2\sigma^2}{2MB_a} - \frac{\eta}{4}p_0 E(\|\nabla F(w_k)\|_2^2) \tag{11}$$

$$\overset{(e)}{\leq} E(F(w_k)) - \eta c(1-\gamma)E(F(w_k) - F^*) - \frac{\eta c p_0}{2}E(F(w_k) - F^*) + \frac{L\eta^2\sigma^2}{2MB_a}$$

$$= E(F(w_k)) - \eta c\gamma' E(F(w_k) - F^*) + \frac{L\eta^2\sigma^2}{2MB_a},$$

where $\gamma' = 1 - \gamma + \frac{p_0}{2}$. Here step (d) follows from Lemma 3, step (e) follows from $F(w)$ is strongly convex with parameter $c$.

Therefore,

$$E(F(w_{k+1}) - F^*) \leq \frac{\eta L\sigma^2}{2c\gamma' MB_a} + (1 - \eta\gamma'c)^{k+1}\left(E(F(w_0) - F^*) - \frac{\eta L\sigma^2}{2c\gamma' MB_a}\right).$$

$$\square$$

*proof of Corollary 1.* Based on $\zeta E(\|\nabla F(w_k) - \nabla F(w_{\tau(m,k)})\|_2^2) \leq \gamma E(\|\nabla F(w_k)\|_2^2)$, the step (b) of (10) should be written as follows,

$$E(F(w_{k+1})) \leq E(F(w_k)) - \frac{\eta}{2}E(\|\nabla F(w_k)\|_2^2) - \frac{\eta}{2M}\sum_{i=1}^{M}E(\|\nabla F(w_{\tau(i,k)})\|_2^2)$$

$$+ \frac{L\eta^2}{2}E(\|v_k\|_2^2) + \frac{\eta}{2M}\sum_{i=1}^{M}E\left(\|\nabla F(w_{\tau(i,k)}) - \nabla F(w_k) \mid \varsigma = 1\|_2^2\right)$$

$$+ \frac{\eta}{2M}\sum_{i=1}^{M}E\left(\|\nabla F(w_{\tau(i,k)}) - \nabla F(w_k) \mid \varsigma \neq 1\|_2^2\right)$$

$$\leq E(F(w_k)) - \frac{\eta}{2}E(\|\nabla F(w_k)\|_2^2) - \frac{\eta}{2M}\sum_{i=1}^{M}E(\|\nabla F(w_{\tau(i,k)})\|_2^2)$$

$$+ \frac{L\eta^2}{2}E(\|v_k\|_2^2) + \frac{\eta\gamma p_1}{2M}\sum_{i=1}^{M}E(\|\nabla F(w_k)\|_2^2) + \frac{\eta\zeta\gamma(1-p_1)}{2M}\sum_{i=1}^{M}E(\|\nabla F(w_k)\|_2^2) \tag{12}$$

$$= E(F(w_k)) - \frac{\eta}{2}\left(1 - p_1\gamma - (1-p_1)\zeta\gamma\right)E(\|\nabla F(w_k)\|_2^2)$$

$$- \frac{\eta}{2M}\sum_{i=1}^{M}E(\|\nabla F(w_{\tau(i,k)})\|_2^2) + \frac{L\eta^2}{2}E(\|v_k\|_2^2)$$

$$\leq E(F(w_k)) - \frac{\eta}{2}\left(1 - p_1\gamma - (1-p_1)\zeta\gamma\right)E(\|\nabla F(w_k)\|_2^2) + \frac{L\eta^2\sigma^2}{2MB_a}$$

$$- \frac{\eta}{4M}\sum_{i=1}^{M}E(\|\nabla F(w_{\tau(i,k)})\|_2^2)$$

$$\leq E(F(w_k)) - \frac{\eta}{2}\left(1 - p_1\gamma - (1-p_1)\zeta\gamma + \frac{p_0}{2}\right)E(\|\nabla F(w_k)\|_2^2) + \frac{L\eta^2\sigma^2}{2MB_a}$$

$$\leq E(F(w_k)) - \eta c\rho E(F(w_k) - F^*) + \frac{L\eta^2\sigma^2}{2MB_a}$$

where $\rho = 1 - p_1\gamma - (1-p_1)\zeta\gamma + \frac{p_0}{2}$, $p_1 = P(\varsigma = 1)$. Therefore,

$$E(F(w_{k+1}) - F^*) \leq \frac{\eta L\sigma^2}{2c\rho MB_a} + (1 - \eta\rho c)^{k+1}\left(E(F(w_0) - F^*) - \frac{\eta L\sigma^2}{2c\rho MB_a}\right).$$

$\square$

*proof of Theorem 2.* Let $w_{k+1} = w_k - \eta v_k$, $v_k = \frac{1}{N_s}\sum_{i=1}^{N_s} g(w_{\tau(i,k)})$, we have

$$
\begin{aligned}
F(w_{k+1}) &\leq F(w_k) + (w_{k+1} - w_k)^T \nabla F(w_k) + \frac{L}{2}\|w_{k+1} - w_k\|_2^2 \\
&= F(w_k) + \langle -\eta v_k, \nabla F(w_k)\rangle + \frac{L}{2}\eta^2\|v_k\|_2^2 \\
&= F(w_k) - \frac{\eta}{N_s}\sum_{i=1}^{N_s}\langle g(w_{\tau(i,k)}), \nabla F(w_k)\rangle + \frac{L}{2}\eta^2\|v_k\|_2^2 \\
&= F(w_k) - \frac{\eta}{N_s}\sum_{i=1}^{N_s}\left(\frac{1}{2}\|g(w_{\tau(i,k)})\|_2^2 + \frac{1}{2}\|\nabla F(w_k)\|_2^2 - \frac{1}{2}\|g(w_{\tau(i,k)}) - \nabla F(w_k)\|_2^2\right) + \frac{L}{2}\eta^2\|v_k\|_2^2 \\
&= F(w_k) - \frac{\eta}{2}\|\nabla F(w_k)\|_2^2 - \frac{\eta}{2N_s}\sum_{i=1}^{N_s}\|g(w_{\tau(i,k)})\|_2^2 + \frac{\eta}{2N_s}\sum_{i=1}^{N_s}\|g(w_{\tau(i,k)}) - \nabla F(w_k)\|_2^2 + \frac{L}{2}\eta^2\|v_k\|_2^2.
\end{aligned}
\tag{13}
$$

Taking expectation,

$$
\begin{aligned}
E(F(w_{k+1})) &\leq E(F(w_k)) - \frac{\eta}{2}E(\|\nabla F(w_k)\|_2^2) - \frac{\eta}{2N_s}\sum_{i=1}^{N_s}E(\|g(w_{\tau(i,k)})\|_2^2) + \frac{L}{2}\eta^2 E(\|v_k\|_2^2) \\
&\quad + \frac{\eta}{2N_s}\sum_{i=1}^{N_s}E\Big(\|g(w_{\tau(i,k)}) - \nabla F(w_k)\|_2^2\Big).
\end{aligned}
\tag{14}
$$

The last term in (14) could be obtained on the basis of $E(g(w_{\tau(i,k)})) = E(\nabla F(w_k))$,

$$
\begin{aligned}
&E\Big(\|g(w_{\tau(i,k)}) - \nabla F(w_k)\|_2^2\Big) \\
&= E(\|g(w_{\tau(i,k)})\|_2^2) + E(\|\nabla F(w_k)\|_2^2) - 2E(\langle g(w_{\tau(i,k)}), \nabla F(w_k)\rangle) \\
&= E(\|g(w_{\tau(i,k)})\|_2^2) - E(\|\nabla F(w_k)\|_2^2).
\end{aligned}
\tag{15}
$$

Returning to (14), we have

$$
\begin{aligned}
E(F(w_{k+1})) &\leq E(F(w_k)) - \frac{\eta}{2}E(\|\nabla F(w_k)\|_2^2) - \frac{\eta}{2N_s}\sum_{i=1}^{N_s}E(\|g(w_{\tau(i,k)})\|_2^2) + \frac{L}{2}\eta^2 E(\|v_k\|_2^2) \\
&\quad + \frac{\eta}{2N_s}\sum_{i=1}^{N_s}E(\|g(w_{\tau(i,k)})\|_2^2) - \frac{\eta}{2N_s}\sum_{i=1}^{N_s}E(\|\nabla F(w_k)\|_2^2).
\end{aligned}
\tag{16}
$$

Similar to Lemma 2,

$$
\begin{aligned}
E(\|v_k\|_2^2) &= E\left(\|\frac{1}{N_s}\sum_{i=1}^{N_s} g(w_{\tau(i,k)})\|_2^2\right) \\
&= \frac{1}{N_s^2}E\left(\|\sum_{i=1}^{N_s}(g(w_{\tau(i,k)}) - \nabla F(w_k))\|_2^2\right) + \frac{1}{N_s^2}E\left(\|\sum_{i=1}^{N_s}\nabla F(w_k)\|_2^2\right) \\
&= \frac{1}{N_s^2}\sum_{i=1}^{N_s}E\Big(\|g(w_{\tau(i,k)}) - \nabla F(w_k)\|_2^2\Big) + E(\|\nabla F(w_k)\|_2^2) \\
&\leq \frac{\sigma^2}{N_s B_s} + \left(\frac{\Theta}{N_s B_s} + 1\right)E(\|\nabla F(w_k)\|_2^2).
\end{aligned}
\tag{17}
$$

Based on (16) and (17), we obtain

$$
\begin{aligned}
E(F(w_{k+1})) &\leq E(F(w_k)) - \eta E(\|\nabla F(w_k)\|_2^2) + \frac{L\eta^2\sigma^2}{2N_sB_s} + \frac{L\eta^2}{2}\left(\frac{\Theta}{N_sB_s} + 1\right)E(\|\nabla F(w_k)\|_2^2) \\
&= E(F(w_k)) - \eta\left(1 - \frac{L\eta\Theta}{2N_sB_s} - \frac{L\eta}{2}\right)E(\|\nabla F(w_k)\|_2^2) + \frac{L\eta^2\sigma^2}{2N_sB_s} \\
&\overset{(a)}{\leq} E(F(w_k)) - \frac{\eta}{2}E(\|\nabla F(w_k)\|_2^2) + \frac{L\eta^2\sigma^2}{2N_sB_s} \\
&\overset{(b)}{\leq} E(F(w_k)) - \eta c E(F(w_k) - F^*) + \frac{L\eta^2\sigma^2}{2N_sB_s}.
\end{aligned}
\tag{18}
$$

Here step (a) follows from $\eta \leq \frac{1}{2L(\frac{\Theta}{N_sB_s}+1)}$, step (b) follows from $2c(F(w) - F^*) \leq \|\nabla F(w)\|_2^2$.

Therefore,

$$
\begin{aligned}
E(F(w_{k+1}) - F^*) &\leq (1 - \eta c)E(F(w_k) - F^*) + \frac{L\eta^2\sigma^2}{2N_sB_s} \\
&\leq (1 - \eta c)^{k+1}\left(E(F(w_0) - F^*) - \frac{L\eta\sigma^2}{2cN_sB_s}\right) + \frac{L\eta\sigma^2}{2cN_sB_s}.
\end{aligned}
\tag{19}
$$

$\square$

## B  Workflows of GBA

In this section, we abstract the workflows of GBA. Algorithm 1 summarizes the workflow on workers in GBA. We implement pipeline between data downloading and data ingestion to accelerate the training. After completing the computation of gradients, the worker would directly send the gradient with the token back to the PS in a non-blocking way. In this way, the fast workers would ingest much more data than the straggling workers. When a worker recovered from a failure, it would drop the previous state (e.g., data in the batch buffer and token) and proceed to deal with the new batch. The disappearance of a specific token would not change the correctness and efficiency of GBA.

---

**Algorithm 1** Workflow on workers

---

**Ensure:** Downloading threads: Download data asynchronously to a *download buffer*;
    Pack threads: Prepare the batch of data from the download buffer to a *batch buffer*;
    Computation threads: Execute the forward and backward pass of the model;
    Communication threads: Pull and push parameters by GRPC;
  1: **Downloading threads**
  2: **repeat**
  3:    Get the addresses of a number of batches (data shard) from PS.
  4:    **repeat**
  5:      **if** The download buffer is not full **then**
  6:        Download a batch of data.
  7:      **else**
  8:        Sleep 100ms.
  9:      **end if**
10:    **until** All the data from this shard has been downloaded
11: **until** No more data shards to get
12:
13: **Computation threads**
14: **repeat**
15:    Get a batch from the batch buffer in a blocking way.
16:    Pull the parameters and fetch a token from PS.
17:    Compute the forward and backward pass.
18:    Send the local gradient and the token back to PS in a non-blocking way.
19: **until** No more data to ingest

---

Algorithm 2 summarizes the workflow on PSs in GBA. The token generation threads, the pull responding threads, and the push responding threads work asynchronously to avoid blocking of the

process. Different from the dense parameters, the embedding parameters are processed and updated by IDs, instead of by the entire embedding parameters. In practice, we optimize the memory usage as we could execute the weighted sum over some gradients in advance based on the tokens.

---

**Algorithm 2** Workflow on PSs

---

**Ensure:** Token generation thread: Generate tokens to the token list;
    Pull responding threads: Send the parameters and a token to the worker;
    Push responding threads: Receive the gradients from a worker and apply them if necessary.
1: **Token generation thread** (Only on PS 0, with lock)
2: **if** Successfully acquire the lock **then**
3:    **if** The number of tokens in the token list is less than the number of workers **then**
4:       Insert new tokens to the tail of the token list (a Queue)
5:    **end if**
6: **end if**
7:
8: **Pull responding threads**
9: Receive the request from a worker with the embedding IDs.
10: Look up the embedding tables based on the IDs for the embedding parameters.
11: Fetch a token from the token list and trigger the token generation thread (only on PS 0).
12: Send the parameters (and the token) back to the worker.
13:
14: **Push responding threads**
15: Receive the gradients from a worker.
16: Store the gradients with the token to the gradient buffer.
17: **if** At least $N_a$ gradients are cached in the gradient buffer **then**
18:    Pop $N_a$ gradients from the gradient buffer.
19:    Update the global step of appearance tagged to each ID.
20:    Decay the gradients of the dense parameters based on the current global step and the attached token.
21:    Decay the gradients of the embedding parameter based on the tagged global step of each ID and the attached token to the gradient.
22:    Calculate the weighted sum of the gradients of the dense parameters, divided by $N_a$.
23:    Calculate the weighted sum of the gradients of the embedding parameters, divided by the number of workers that encountered the particular ID.
24: **end if**

---

## C Detailed statistics of Table 6 in the submission

Here we want to clarify the statements about Figure 6 in the submission, and present the detailed statistics. Table .1-.3 depict the AUCs after inheriting the checkpoints trained via synchronous training. Table .5-.7 introduce the AUCs after inheriting the checkpoints trained by the compared training modes and being switched to synchronous training. We collect the mean AUCs from the first day, the last day, and all days across the three datasets, as shown in Table .4 and Table .8. We can infer from the two tables that GBA provides the closest AUC scores as synchronous training. GBA appears with the lowest immediate AUC drop after switching, i.e., merely 0.1% decrement after switching from/to synchronous training at the first day. Throughout the three datasets, GBA outperforms the other baselines by at least 0.2% (Hop-BW in Table .4) when switching from synchronous training and 0.1% (Hop-BS in Table .8) when switching to synchronous training.

## D Proof of sudden drop in performance after switching.

**Theorem 3.** *Based on the above Assumption and* $\frac{1}{NL(\frac{\Theta}{B}+1)} \leq \eta \leq \frac{1}{2L(\frac{\Theta}{B}+1)}$, *also suppose that for some* $\gamma \leq 1$,

$$E\Big(\|\nabla F(w_k) - \nabla F(w_{\tau(i,k)})\|_2^2\Big) \leq \gamma E(\|\nabla F(w_k)\|_2^2),$$

*in the asynchronous training, the expectation of loss in the* $k+1$ *step is deduced by*

$$E(F(w_{k+1})) \leq E(F(w_k)) - \frac{\eta}{2}(1-\gamma)E(\|\nabla F(w_k)\|_2^2) + \frac{L\eta^2\sigma^2}{2B} - \frac{\eta}{4}E(\|\nabla F(w_{\tau(k)})\|_2^2). \quad (20)$$

Table .1: Figure 6(a) - Criteo (from Sync.)

| Date | Sync. | GBA | Hop-BW | Hop-BS | BSP | Aysnc. |
|------|-------|-----|--------|--------|-----|--------|
| 13 | 0.7999 | 0.7964 | 0.7954 | 0.7924 | 0.7930 | 0.5000 |
| 14 | 0.7957 | 0.7932 | 0.7959 | 0.7869 | 0.7886 | 0.5000 |
| 15 | 0.7967 | 0.7957 | 0.7895 | 0.7891 | 0.7889 | 0.5000 |
| 16 | 0.7963 | 0.7956 | 0.7932 | 0.5040 | 0.7891 | 0.5000 |
| 17 | 0.7962 | 0.7955 | 0.7930 | 0.5040 | 0.7883 | 0.5000 |
| 18 | 0.7957 | 0.7950 | 0.7939 | 0.5030 | 0.7862 | 0.5000 |
| 19 | 0.7972 | 0.7966 | 0.7968 | 0.5030 | 0.7863 | 0.5000 |
| 20 | 0.7974 | 0.7973 | 0.7985 | 0.5030 | 0.7868 | 0.5000 |
| 21 | 0.7965 | 0.7959 | 0.7948 | 0.5050 | 0.7863 | 0.5000 |
| 22 | 0.7957 | 0.7955 | 0.7939 | 0.5040 | 0.7865 | 0.5000 |
| 23 | 0.7987 | 0.7986 | 0.7933 | 0.5060 | 0.7871 | 0.5000 |
| Avg. | 0.7969 | 0.7959 | 0.7944 | 0.5819 | 0.7879 | 0.5000 |

Table .2: Figure 6(b) - Alimama (from Sync.)

| Date | Sync. | GBA | Hop-BW | Hop-BS | BSP | Aysnc. |
|------|-------|-----|--------|--------|-----|--------|
| 6 | 0.6490 | 0.6489 | 0.6472 | 0.6488 | 0.6472 | 0.5000 |
| 7 | 0.6503 | 0.6502 | 0.6478 | 0.6503 | 0.6500 | 0.5000 |
| 8 | 0.6523 | 0.6523 | 0.6483 | 0.6523 | 0.6512 | 0.5000 |
| Avg. | 0.6505 | 0.6505 | 0.6478 | 0.6505 | 0.6495 | 0.5000 |

Table .3: Figure 6(c) - Private (from Sync.)

| Date | Sync. | GBA | Hop-BW | Hop-BS | BSP | Aysnc. |
|------|-------|-----|--------|--------|-----|--------|
| 15 | 0.7877 | 0.7880 | 0.7870 | 0.7875 | 0.7880 | 0.7795 |
| 16 | 0.7874 | 0.7877 | 0.7860 | 0.7878 | 0.7870 | 0.7785 |
| 17 | 0.7856 | 0.7860 | 0.7840 | 0.7858 | 0.7850 | 0.7774 |
| 18 | 0.7884 | 0.7888 | 0.7850 | 0.7882 | 0.7877 | 0.7873 |
| 19 | 0.7894 | 0.7905 | 0.7855 | 0.7890 | 0.7886 | 0.7878 |
| 20 | 0.7785 | 0.7788 | 0.7750 | 0.7781 | 0.7774 | 0.7598 |
| 21 | 0.7865 | 0.7868 | 0.7823 | 0.7863 | 0.7850 | 0.7769 |
| 22 | 0.7862 | 0.7870 | 0.7825 | 0.7858 | 0.7862 | 0.7754 |
| Avg. | 0.7862 | 0.7867 | 0.7834 | 0.7861 | 0.7856 | 0.7778 |

Table .4: Average AUC decrement on three datasets between GBA and the other baselines (from Sync.)

| | Sync. | Hop-BW | Hop-BS | BSP | Aysnc. |
|---|-------|--------|--------|-----|--------|
| 1st day | +0.0011 | -0.0012 | -0.0015 | -0.0017 | -0.1513 |
| last day | -0.0002 | -0.0046 | -0.0979 | -0.0045 | -0.1542 |
| Average | +0.0002 | -0.0025 | -0.0716 | -0.0034 | -0.1518 |

Table .5: Figure 6(d) - Criteo (to Sync.)

| Date | Sync. | GBA | Hop-BW | Hop-BS | BSP | Aysnc. |
|---|---|---|---|---|---|---|
| 13 | 0.7999 | 0.7963 | 0.7968 | 0.7937 | 0.7913 | 0.7872 |
| 14 | 0.7957 | 0.7947 | 0.7933 | 0.7917 | 0.7907 | 0.7902 |
| 15 | 0.7967 | 0.7957 | 0.7952 | 0.7932 | 0.7923 | 0.7922 |
| 16 | 0.7963 | 0.7954 | 0.7956 | 0.7937 | 0.7944 | 0.7939 |
| 17 | 0.7962 | 0.7956 | 0.7946 | 0.7943 | 0.7945 | 0.7935 |
| 18 | 0.7957 | 0.7958 | 0.7949 | 0.7931 | 0.7922 | 0.7929 |
| 19 | 0.7972 | 0.7966 | 0.7960 | 0.7952 | 0.7944 | 0.7946 |
| 20 | 0.7974 | 0.7970 | 0.7970 | 0.7962 | 0.7957 | 0.7957 |
| 21 | 0.7965 | 0.7963 | 0.7956 | 0.7950 | 0.7931 | 0.7952 |
| 22 | 0.7957 | 0.7955 | 0.7955 | 0.7953 | 0.7940 | 0.7950 |
| 23 | 0.7987 | 0.7982 | 0.7978 | 0.7973 | 0.7964 | 0.7972 |
| Avg. | 0.7969 | 0.7961 | 0.7957 | 0.7944 | 0.7935 | 0.7934 |

Table .6: Figure 6(e) - Alimama (to Sync.)

| Date | Sync. | GBA | Hop-BW | Hop-BS | BSP | Aysnc. |
|---|---|---|---|---|---|---|
| 6 | 0.6490 | 0.6492 | 0.6401 | 0.6484 | 0.6452 | 0.6352 |
| 7 | 0.6503 | 0.6504 | 0.6437 | 0.6499 | 0.6487 | 0.6426 |
| 8 | 0.6523 | 0.6523 | 0.6471 | 0.6520 | 0.6503 | 0.6456 |
| Avg. | 0.6505 | 0.6506 | 0.6436 | 0.6501 | 0.6481 | 0.6411 |

Table .7: Figure 6(f) - Private (to Sync.)

| Date | Sync. | GBA | Hop-BW | Hop-BS | BSP | Aysnc. |
|---|---|---|---|---|---|---|
| 15 | 0.7877 | 0.7878 | 0.7783 | 0.7859 | 0.7732 | 0.7870 |
| 16 | 0.7874 | 0.7877 | 0.7844 | 0.7867 | 0.7767 | 0.5000 |
| 17 | 0.7856 | 0.7855 | 0.7813 | 0.7853 | 0.7782 | 0.5000 |
| 18 | 0.7884 | 0.7883 | 0.7858 | 0.7880 | 0.7805 | 0.5000 |
| 19 | 0.7894 | 0.7896 | 0.7870 | 0.7889 | 0.7848 | 0.5000 |
| 20 | 0.7785 | 0.7786 | 0.7761 | 0.7784 | 0.7746 | 0.5000 |
| 21 | 0.7865 | 0.7865 | 0.7843 | 0.7864 | 0.7828 | 0.5000 |
| 22 | 0.7862 | 0.7863 | 0.7855 | 0.7861 | 0.7838 | 0.5000 |
| Avg. | 0.7862 | 0.7863 | 0.7828 | 0.7857 | 0.7793 | 0.5359 |

Table .8: Average AUC decrement on three datasets between GBA and the other baselines (to Sync.)

| | Sync. | Hop-BW | Hop-BS | BSP | Aysnc. |
|---|---|---|---|---|---|
| 1st day | +0.0011 | -0.0060 | -0.0018 | -0.0079 | -0.0080 |
| last day | +0.0001 | -0.0021 | -0.0005 | -0.0021 | -0.0980 |
| Average | +0.0002 | -0.0036 | -0.0009 | -0.0040 | -0.0875 |

*If we switch to the **synchronous training** in the $k+1$ step, the expectation of loss is shown as follows*

$$E(F(w_{k+1})) \le E(F(w_k)) - \frac{\eta}{2}(1-\gamma)E(\|\nabla F(w_k)\|_2^2) + \frac{L\eta^2\sigma^2}{2B} + \frac{L\eta^2\sigma^2}{2BN}$$
$$+ (\frac{L\eta^2\Theta}{2B} + \frac{L\eta^2}{2} - \frac{\eta}{2N})E\left(\|\nabla F(w_k)\|_2^2\right) - \frac{\eta}{4N}E\left(\|\nabla F(w_{\tau(k)})\|_2^2\right) \tag{21}$$

*and switching the asynchronous mode to the synchronous mode may drop in performance.*

*proof of Theorem 3.* According to Theorem 1, (20) could be obtained if we apply the asynchronous training mode. Next, we will prove (21).

Let $w_{k+1} = w_k - \eta v_k$, $v_k = \frac{1}{N}\sum_{i=1}^{N} g(w_{\tau(i,k)})$, we have

$$
\begin{aligned}
F(w_{k+1}) &\le F(w_k) + (w_{k+1} - w_k)^T\nabla F(w_k) + \frac{L}{2}\|w_{k+1} - w_k\|_2^2 \\
&\le F(w_k) + \langle -\eta v_k, \nabla F(w_k)\rangle + \frac{L}{2}\eta^2\|v_k\|_2^2 \\
&= F(w_k) - \frac{\eta}{N}\sum_{i=1}^{N}\langle g(w_{\tau(i,k)}), \nabla F(w_k)\rangle + \frac{L}{2}\eta^2\|v_k\|_2^2.
\end{aligned}
\tag{22}
$$

Since we switch the asynchronous training to the synchronous training, the gradient of each worker $g(w_{\tau(i,k)})$, $i = 1, 2, ..., N$, in the $k$ step may be different.

Owing to $2\langle x, y\rangle = \|x\|_2^2 + \|y\|_2^2 - \|x - y\|_2^2$, the expectation of (22) is shown as follows,

$$
\begin{aligned}
E(F(w_{k+1})) &\le E(F(w_k)) - \frac{\eta}{2}E(\|\nabla F(w_k)\|_2^2) - \frac{\eta}{2N}\sum_{i=1}^{N}E(\|g(w_{\tau(i,k)})\|_2^2) \\
&+ \frac{\eta}{2N}\sum_{i=1}^{N}E\left(\|g(w_{\tau(i,k)}) - \nabla F(w_k)\|_2^2\right) + \frac{L}{2}\eta^2E(\|v_k\|_2^2).
\end{aligned}
\tag{23}
$$

Since there applied a gradient in the $k$ step of the asynchronous training, we may assume $g(w_{\tau(1,k)}) = g(w_k)$. Hence,

$$
\begin{aligned}
&\frac{\eta}{2N}\sum_{i=1}^{N}E\left(\|g(w_{\tau(i,k)}) - \nabla F(w_k)\|_2^2\right) \\
&= \frac{\eta}{2N}\left(E(\|g(w_{\tau(1,k)}) - \nabla F(w_k)\|_2^2) + \sum_{j=2}^{N}E(\|g(w_{\tau(j,k)}) - \nabla F(w_k)\|_2^2)\right) \\
&= \frac{\eta}{2N}\left(E(\|g(w_k)\|_2^2) - E(\|\nabla F(w_k)\|_2^2)\right) \\
&+ \frac{\eta}{2N}\sum_{j=2}^{N}\left(E(\|g(w_{\tau(j,k)})\|_2^2) - E(\|\nabla F(w_{\tau(j,k)})\|_2^2) + E(\|\nabla F(w_{\tau(j,k)}) - \nabla F(w_k)\|_2^2)\right)
\end{aligned}
\tag{24}
$$

Besides,

$$\frac{\eta}{2N}\sum_{i=1}^{N}E(\|g(w_{\tau(i,k)})\|_2^2) = \frac{\eta}{2N}E(\|g(w_k)\|_2^2) + \frac{\eta}{2N}\sum_{j=2}^{N}E(\|g(w_{\tau(j,k)})\|_2^2). \tag{25}$$

Based on (24) and (25), we have

$$E(F(w_{k+1})) \le E(F(w_k)) - \frac{\eta}{2}E(\|\nabla F(w_k)\|_2^2) - \frac{\eta}{2N}E(\|\nabla F(w_k)\|_2^2) + \frac{L}{2}\eta^2 E(\|v_k\|_2^2)$$

$$- \frac{\eta}{2N}\sum_{j=2}^N E(\|\nabla F(w_{\tau(j,k)})\|_2^2) + \frac{\eta}{2N}\sum_{j=2}^N E(\|\nabla F(w_{\tau(j,k)}) - \nabla F(w_k)\|_2^2)$$

$$\overset{(a)}{\le} E(F(w_k)) - \frac{\eta}{2}E(\|\nabla F(w_k)\|_2^2) - \frac{\eta}{2N}E(\|\nabla F(w_k)\|_2^2) + \frac{L}{2}\eta^2 E(\|v_k\|_2^2)$$

$$- \frac{\eta}{2N}\sum_{j=2}^N E(\|\nabla F(w_{\tau(j,k)})\|_2^2) + \frac{\gamma\eta}{2}E(\|\nabla F(w_k)\|_2^2) \qquad (26)$$

$$= E(F(w_k)) - \frac{\eta}{2}(1-\gamma)E(\|\nabla F(w_k)\|_2^2) - \frac{\eta}{2N}E(\|\nabla F(w_k)\|_2^2) + \frac{L}{2}\eta^2 E(\|v_k\|_2^2)$$

$$- \frac{\eta}{2N}\sum_{j=2}^N E(\|\nabla F(w_{\tau(j,k)})\|_2^2).$$

Here, step (a) follows from $E\Big(\|\nabla F(w_k) - \nabla F(w_{\tau(j,k)})\|_2^2\Big) \le \gamma E(\|\nabla F(w_k)\|_2^2).$

$$E(\|v_k\|_2^2) = E\Big(\|\frac{1}{N}\sum_{i=1}^N g(w_{\tau(i,k)})\|_2^2\Big) = \frac{1}{N^2}E\Big(\|\sum_{i=1}^N g(w_{\tau(i,k)})\|_2^2\Big)$$

$$= \frac{1}{N^2}E\Big(\|\sum_{i=1}^N g(w_{\tau(i,k)}) - \nabla F(w_{\tau(i,k)}) + \nabla F(w_{\tau(i,k)})\|_2^2\Big)$$

$$= \frac{1}{N^2}E\Big(\|\sum_{i=1}^N g(w_{\tau(i,k)}) - \nabla F(w_{\tau(i,k)})\|_2^2\Big) + \frac{1}{N^2}E\Big(\|\sum_{i=1}^N \nabla F(w_{\tau(i,k)})\|_2^2\Big)$$

$$\le E\Big(\|g(w_k) - \nabla F(w_k)\|_2^2\Big) + \frac{1}{N^2}\sum_{j=2}^N E\Big(\|g(w_{\tau(j,k)}) - \nabla F(w_{\tau(j,k)})\|_2^2\Big) \qquad (27)$$

$$+ E\Big(\|\nabla F(w_k)\|_2^2\Big) + \frac{1}{N}\sum_{j=2}^N E\Big(\|\nabla F(w_{\tau(j,k)})\|_2^2\Big)$$

$$\le \frac{\sigma^2}{B} + \frac{\Theta}{B}E\Big(\|\nabla F(w_k)\|_2^2\Big) + \frac{1}{N^2}\sum_{j=2}^N \Big(\frac{\sigma^2}{B} + \frac{\Theta}{B}E(\|\nabla F(w_{\tau(j,k)})\|_2^2)\Big)$$

$$+ E\Big(\|\nabla F(w_k)\|_2^2\Big) + \frac{1}{N}\sum_{j=2}^N E\Big(\|\nabla F(w_{\tau(j,k)})\|_2^2\Big)$$

Based on (27), we obtain

$$E(F(w_{k+1})) \le E(F(w_k)) - \frac{\eta}{2}(1-\gamma)E(\|\nabla F(w_k)\|_2^2) + \frac{L\eta^2\sigma^2}{2B} + \frac{L\eta^2\sigma^2}{2BN}$$

$$+ \Big(\frac{L\eta^2\Theta}{2B} + \frac{L\eta^2}{2} - \frac{\eta}{2N}\Big)E\Big(\|\nabla F(w_k)\|_2^2\Big) \qquad (28)$$

$$+ \sum_{j=2}^N \Big(\frac{L\eta^2\Theta}{2BN^2} + \frac{L\eta^2}{2N} - \frac{\eta}{2N}\Big)E\Big(\|\nabla F(w_{\tau(j,k)})\|_2^2\Big).$$

In (28),

$$\sum_{j=2}^N \Big(\frac{L\eta^2\Theta}{2BN^2} + \frac{L\eta^2}{2N} - \frac{\eta}{2N}\Big)E\Big(\|\nabla F(w_{\tau(j,k)})\|_2^2\Big)$$

$$\qquad (29)$$

$$= \frac{\eta}{2N}\sum_{j=2}^N \Big(\frac{L\eta\Theta}{BN} + L\eta - 1\Big)E\Big(\|\nabla F(w_{\tau(j,k)})\|_2^2\Big) \le -\frac{\eta}{4N}E\Big(\|\nabla F(w_{\tau(k)})\|_2^2\Big).$$

Therefore,

$$E(F(w_{k+1})) \le E(F(w_k)) - \frac{\eta}{2}(1-\gamma)E(\|\nabla F(w_k)\|_2^2) + \frac{L\eta^2\sigma^2}{2B} + \frac{L\eta^2\sigma^2}{2BN}$$

$$+ \left(\frac{L\eta^2\Theta}{2B} + \frac{L\eta^2}{2} - \frac{\eta}{2N}\right)E\left(\|\nabla F(w_k)\|_2^2\right) - \frac{\eta}{4N}E\left(\|\nabla F(w_{\tau(k)})\|_2^2\right) \tag{30}$$

Owing to $\eta \ge \frac{1}{NL(\frac{\Theta}{B}+1)}$, we have $\frac{L\Theta\eta}{B} + L\eta \ge \frac{1}{N}$ and

$$\left(\frac{L\eta^2\Theta}{2B} + \frac{L\eta^2}{2} - \frac{\eta}{2N}\right)E\left(\|\nabla F(w_k)\|_2^2\right) = \frac{\eta}{2}\left(\frac{L\eta\Theta}{B} + L\eta - \frac{1}{N}\right)E\left(\|\nabla F(w_k)\|_2^2\right) \ge 0. \tag{31}$$

Since $-\frac{\eta}{4N}E\left(\|\nabla F(w_{\tau(k)})\|_2^2\right) > -\frac{\eta}{4}E\left(\|\nabla F(w_{\tau(k)})\|_2^2\right)$, we have

$$E(F(w_k)) - \frac{\eta}{2}(1-\gamma)E(\|\nabla F(w_k)\|_2^2) + \frac{L\eta^2\sigma^2}{2B} - \frac{\eta}{4}E(\|\nabla F(w_{\tau(k)})\|_2^2)$$

$$\le E(F(w_k)) - \frac{\eta}{2}(1-\gamma)E(\|\nabla F(w_k)\|_2^2) + \frac{L\eta^2\sigma^2}{2B} + \frac{L\eta^2\sigma^2}{2BN}$$

$$+ \left(\frac{L\eta^2\Theta}{2B} + \frac{L\eta^2}{2} - \frac{\eta}{2N}\right)E\left(\|\nabla F(w_k)\|_2^2\right) - \frac{\eta}{4N}E\left(\|\nabla F(w_{\tau(k)})\|_2^2\right). \tag{32}$$

Therefore, switching the asynchronous mode to the synchronous mode in the $k+1$ step may drop in performance. $\qquad\square$

**Theorem 4.** *Based on the above Assumption and* $\eta \le \frac{1}{2L(\frac{\Theta}{B}+1)}$, *__in the synchronous training__, the expectation of loss in the $k+1$ step is deduced by*

$$E(F(w_{k+1})) \le E(F(w_k)) - \eta\left(1 - \frac{L\eta\Theta}{2NB} - \frac{L\eta}{2}\right)E(\|\nabla F(w_k)\|_2^2) + \frac{L\eta^2\sigma^2}{2NB}. \tag{33}$$

*If we switch to the __asynchronous training__ in the $k+1$ step, the expectation of loss becomes as follows*

$$E(F(w_{k+1})) \le E(F(w_k)) - \eta(1 - \frac{L\eta\Theta}{2B} - \frac{L\eta}{2})E(\|\nabla F(w_k))\|_2^2 + \frac{L\eta^2\sigma^2}{2B}, \tag{34}$$

*and switching the synchronous mode to the asynchronous mode may drop in performance.*

*proof of Theorem 4.* According to the Theorem 2, (33) could be obtained. Next, we will prove (34).

$$E(F(w_{k+1})) \le E(F(w_k)) - \frac{\eta}{2}E(\|\nabla F(w_k\|_2^2) - \frac{\eta}{2}E(\|g(w_{\tau(1,k)})\|_2^2)$$

$$+ \frac{\eta}{2}E\left(\|g(w_{\tau(1,k)}) - \nabla F(w_k)\|_2^2\right) + \frac{L}{2}\eta^2 E(\|v_k\|_2^2). \tag{35}$$

Since $E\left(\|g(w_{\tau(1,k)}) - \nabla F(w_k)\|_2^2\right) = E(\|g(w_k)\|_2^2) - E(\|\nabla F(w_k)\|_2^2)$, we have

$$E(F(w_{k+1})) \le E(F(w_k)) - \eta E(\|\nabla F(w_k)\|_2^2) + \frac{L\eta^2}{2}E(\|v_k\|_2^2).$$

Owing to

$$E(\|v_k\|_2^2) = E(\|g(w_{\tau(1,k)})\|_2^2) = E(\|g(w_{\tau(1,k)}) - \nabla F(w_k) + \nabla F(w_k)\|_2^2)$$

$$= E(\|g(w_{\tau(1,k)}) - \nabla F(w_k))\|_2^2) + E(\|\nabla F(w_k))\|_2^2) \tag{36}$$

$$\le \frac{\sigma^2}{B} + (\frac{\Theta}{B} + 1)E(\|\nabla F(w_k)\|_2^2),$$

we obtain

$$E(F(w_{k+1})) \le E(F(w_k)) - \eta(1 - \frac{L\eta}{2} - \frac{L\eta\Theta}{2B})E(\|\nabla F(w_k))\|_2^2 + \frac{L\eta^2\sigma^2}{2B}. \tag{37}$$

Owing to $\frac{L\eta^2\sigma^2}{2B} > \frac{L\eta^2\sigma^2}{2NB}$ and $\frac{L\eta\Theta}{2B} > \frac{L\eta\Theta}{2NB}$, we have

$$E(F(w_k)) - \eta\left(1 - \frac{L\eta\Theta}{2NB} - \frac{L\eta}{2}\right)E(\|\nabla F(w_k)\|_2^2) + \frac{L\eta^2\sigma^2}{2NB}$$

$$\le E(F(w_k)) - \eta(1 - \frac{L\eta\Theta}{2B} - \frac{L\eta}{2})E(\|\nabla F(w_k))\|_2^2 + \frac{L\eta^2\sigma^2}{2B}. \tag{38}$$

Hence, switching the synchronous mode to the asynchronous mode may drop in performance. $\qquad\square$