# OpenReview forum: "GBA: A Tuning-free Approach to Switch between Synchronous and Asynchronous Training for Recommendation Models"
_NeurIPS.cc/2022/Conference — NeurIPS 2022 Accept_

### Official Review · Reviewer_pSV7 · 2022-07-07

**Rating:** 7
**Confidence:** 3
**Ethics Flag:** Yes
**Soundness:** 3 good
**Presentation:** 4 excellent
**Contribution:** 3 good

**Summary:**

The authors of the paper "GBA: A Tuning-free Approach to Switch between Synchronous and Asynchronous Training for Recommendation Models" propose a new training method for the recommender systems that enables switching between two different modes: synchronous and asynchronous training benefiting from both approaches and allowing for continuous learning maintaining the same accuracy and efficiency without hyperparameter tuning. Users of such a system can select a suitable training mode according to their judgment of cluster workload.

**Questions:**

I have only a couple of questions:
* In Section 5.2 you write that "We can see
261 that the low gradient staleness helps Hop-BS receive a better accuracy than BSP" (line 260-261). When I look at Figure 6 (a-c), I observe that HOP-BS performs the worst in the (a) setting dropping right after the beginning (Test day 14) while in other settings (b,c) it is not possible to distinguish any difference in AUC performance. BSP (dark blue) is performing clearly better. Next, you write that HOP-BW performs the worst, while I observe a relatively normal performance. I assume HOP-BS and HOP-BW got mixed up here. However, it doesn't explain that you say that GBA improves the AUC on (a) and (c) by at least 0.2% but I can't see this improvement in (a) and I don't think on (c) it gets to 0.2% difference. I would suggest double-checking this part as it confuses readers at this point.
* In addition, I think Figure 6 description states (d-e) while it is supposed to be (d-f)
* Another suggestion would be to highlight in each of the tables the best-performing metrics as you use quite a few of them and it becomes more difficult to check. This is a minor suggestion but can potentially improve readability.





**Limitations:**

The limitations were not explicitly discussed with the only suggestion for future work to include automatic switching between the training modes. I think the authors might want to consider unpacking the limitations of this work a bit more.

**Strengths And Weaknesses:**

Post Rebuttal comments: After the changes provided by the authors, I've updated my score to accept

------------------------------------------
Originality

The paper solves a problem that, though specific to recommender systems, can be very beneficial in a practical setting and valuable for all the industries attempting to train large recommendation systems under imperfect computing constraints. I would judge that the originality of the paper is high as, according to the authors, nobody has attempted to provide a hyperparameter tuning-free approach to the switching between synchronous and asynchronous training modes.

Quality

The quality of this paper is also high - it provides a clear overview, and outlines several important insights and observations on the performance of the training modes that help to determine and address the gaps in the current work.  This provides clarity on the choices and explains the conducted experiments.

Clarity

The paper is well-structured and well-written. I enjoyed reading it. The only problem that undermines the clarity is Section 5.2 and more specifically, Figure 6. I will outline my confusion in the Questions section below. In summary, it seems that the methods in the figure and their performance do not correspond to the description in the text.

Significance

I find that this work might have a medium impact in addressing important challenges specific to the recommender systems' properties in the industrial setting. I believe that this impact might translate to better resource utilization and, therefore, cost savings.

---

> ### Author Response · Authors · 2022-08-02
> **Response to Reviewer pSV7**
>
> Thank you for the time and constructive feedback. We really appreciate your positive comments about our work. We have submitted an updated version of the paper, and will address your questions as follows.
>
> **Question-1**: About Figure 6.
>
> **Answer to Question-1**: Please refer to our general response attached to this submission at the top of this page, where we provide the detailed statistics and some supplementary discussion to Figure 6 of the submission. In Figure 6, the major concern should be the tendency of the AUC scores throughout training. In particular, we focus on the immediate AUC after switching (i.e., AUC on the first day), the AUC after several day's training (i.e., AUC on the last day), and the average AUC throughout the training data. Table R4 indicates that GBA outperforms the best baseline (i.e., Hop-BW) by about 0.2% after switching from synchronous training. GBA also provides almost the same AUC score on average as well as on the last day compared to synchronous training. Besides, we can infer from Table R4 that the performance of Hop-BW and Hop-BS is not stable after switching, i.e., Hop-BW performs much better on switching from synchronous training than switching to synchronous training, while Hop-BS works only well when switching to synchronous training (Table C.8 in Appendix). We have polished the statement of Figure 6 and fixed the typos in our revised submission.

---

### Official Review · Reviewer_S16x · 2022-07-11

**Rating:** 7
**Confidence:** 4
**Soundness:** 4 excellent
**Presentation:** 4 excellent
**Contribution:** 4 excellent

**Summary:**

The paper proposes a novel algorithm to switch between two common distributed training modes (abbreviated as PS and AR in the paper) without sacrificing accuracy and performance while also not tuning hyperparameters. The paper identifies three key observations that govern performance and accuracy and based on these, proposes a global batch aggregation scheme that mimics the global gradients in synchronous training to switch between PS to AR without any loss in accuracy. The paper shows a convergence analysis of the algorithm and evaluates its performance on three different tasks showing it has better performance and accuracy overall.


**Questions:**

- How does the performance vary with the number of workers? Since it relies on global batch aggregation, there may be some dependence on the number of workers?

**Ethics Review Area:**

["I don’t know"]

**Limitations:**

The authors discussed the limitations of the work in the conclusion section.

**Strengths And Weaknesses:**

Strengths:
- The paper proposes a novel algorithm to effectively switch between two training modes without tuning hyper parameters which is costly.
- he paper theoretically proves convergence.
- The proposed algorithm performs better than state-of-the-art recommender training schemes on three different tasks.

Weakness:
- The paper should provide details on when to initiate switching based on cluster status and ablation studies that show how the model performs for different cluster statuses.

---

> ### Author Response · Authors · 2022-08-02
> **Response to Reviewer S16x**
>
> Thank you for the valuable comments. We really appreciate your positive comments about our work. We will address the issues as follows.
>
> **Weakness-1**: The paper should provide details on when to initiate switching based on cluster status and ablation studies that show how the model performs for different cluster statuses.
>
> **Answer to Weakness-1**: In Section 3.1, we use CPU utilization to measure the cluster status and QPS to show the training performance of aysnc and sync modes for different cluster statuses. Currently, our users can only switch manually based on their experience and requirements for training performance. GBA makes it possible to switch between all-reduce sync mode and PS async mode. The guidelines for automatic switching would be derived from more analyses upon the training trace logs. And it could be formulated as an optimization problem under many control factors, including but not limited to the overall QPS, training cost, and task scheduling with priority. Since it needs deeper study, we consider it as one of our future work.
>
> Furthermore, we collected the performance of GBA under three representative cluster statuses (different time of a day), and have depicted the detailed metrics in Table 3 in Section 5.3. The QPS of the asynchronous training indicates the huge differences among the three time periods of training. From Table 3 we see that GBA appears to have similar training efficiency as the asyn mode, and provides comparable AUC scores to the sync mode. The result also implies that the model accuracy is not sensitive to the switching, which means that it is safe to switch the training mode by users (either manually or automatically in the future).
>
> **Question-1**: How does the performance vary with the number of workers? Since it relies on global batch aggregation, there may be some dependence on the number of workers?
>
> **Answer to Question-1**: In our experiments of Section 5.3, we show that GBA has good scalability. Figure 7 shows that we could decrease the local batch size and increase the worker number, while the AUC scores remain nearly the same. Although the optimization process of training relies on global batch aggregation, the worker number is limited by the physical implementation. It means that we cannot use an extremely large local batch size (incurring the out-of-memory failure) nor too many workers (incurring severe communication bottleneck on PS nodes due to too many connections). Therefore, the experiments in Figure 7 were conducted over four practical numbers of workers which ensures resource efficiency.

---

### Official Review · Reviewer_JyKb · 2022-07-11

**Rating:** 5
**Confidence:** 4
**Soundness:** 2 fair
**Presentation:** 2 fair
**Contribution:** 2 fair

**Summary:**

This paper proposes a mechanism called global batch gradient aggregation (GBA) on parameter server architectures that enables effective switching between synchronous and asynchronous training of deep learning-based recommendation systems.

**Questions:**

Figure 6 is hard to track, would it be possible to list the final AUC of each model for each task in a table?

**Limitations:**

Not applicable.

**Strengths And Weaknesses:**

Strengths:

- The discovery of the phenomenon that directly switching between synchronous and asynchronous training of deep learning-based recommendation models leads to a significant drop in performances is interesting.

Weaknesses:

- The explanation of the phenomenon is lack of technique depth. The insights presented in Section 3.2 are intuitive rather than analytical. It would make the paper stronger if there were some formal theoretical analysis on the drop in performance.

- The performance boost is marginal; for example, in Figure 6, it is very difficult to understand the performance gain w.r.t. the proposed method. To be more concrete, it seems that, for the Criteo dataset, synchronous training achieves the optimal AUC much ealier than GBA.

---

> ### Author Response · Authors · 2022-08-02
> **Response to Reviewer JyKb**
>
> Thank you for your efforts and comments. We have submitted an updated version of the paper, and will respond to your comments in detail below. What we would like to highlight is that both weaknesses you mentioned are outside the scope of our claimed contributions. We would greatly appreciate it if you could reconsider our contributions and revisit the review in light of our clarifications.
>
> **Weakness-1**: The explanation of the phenomenon is lack of technique depth. The insights presented in Section 3.2 are intuitive rather than analytical. It would make the paper stronger if there were some formal theoretical analysis on the drop in performance.
>
> **Answer to Weakness-1**:
> According to your comments, we have added more related work and discussion in Section 3.2 to make it more clear and more convincing (Line 144 [2,17]). In addition, we provided an analysis on the sudden drop in performance after switching. It is challenging to perform a comprehensive theoretical analysis of this phenomenon, which remains an open question in the community. The drop in performance after switching is related to many factors of the implementation, such as batch size and optimizer settings. We analyzed the phenomenon from the perspective of batch size based on our convergence analysis in Section 4.2. The detailed theoretical analysis can be found in Appendix D of the updated version, and we briefly summarize it as follows:
>
> In synchronous training, if we do not switch the training mode, the expectation of loss in the $k+1$ step is
> $$
> E(F(w_{k+1}))\leq E(F(w_{k}))-\eta\left(1-\frac{L\eta\Theta}{2NB}-\frac{L\eta}{2}\right)E(\|\nabla F(w_{k})\|_{2}^{2})+\frac{L\eta^{2}\sigma^{2}}{2NB}.
> $$
>
> However, if we switch to the asynchronous mode, the expectation of loss in the $k+1$ step is
> $$
> E(F(w_{k+1}))\leq E(F(w_{k}))-\eta(1-\frac{L\eta\Theta}{2B}-\frac{L\eta}{2})E(\|\nabla F(w_{k}))\|_{2}^{2}+\frac{L\eta^{2}\sigma^{2}}{2B}.
> $$
> Since $\frac{L\eta^{2}\sigma^{2}}{2B}>\frac{L\eta^{2}\sigma^{2}}{2NB}$ and $\frac{L\eta\Theta}{2B}>\frac{L\eta\Theta}{2NB}$, switching the synchronous mode to the asynchronous mode may drop in performance. A similar conclusion can be found regarding the drop in performance from asynchronous mode to synchronous mode, and we depict the deduction in Appendix D.
>
> However, we would like to highlight that the analysis of this phenomenon is **not the main point/contribution** of this paper. The observation of this phenomenon just motivated us to propose GBA. GBA allows switching the training modes without tuning hyper-parameters, so as to avoid those complex influences caused by the hyper-parameters and avoid the sudden drop in performance.
>
> **Weakness-2**: The performance boost is marginal; for example, in Figure 6, it is very difficult to understand the performance gain w.r.t. the proposed method. To be more concrete, it seems that, for the Criteo dataset, synchronous training achieves the optimal AUC much ealier than GBA.
>
> **Answer to Weakness-2**:
> We present the detailed statistics of Figure 6 in Appendix C (also in the general response at the top of this page). Originally, we expect to emphasize the tendency of the AUC scores after switching in Figure 6. GBA provides the most similar tendency to sync training, i.e., a promising and stable accuracy from the first day to the last day of the dataset. From the perspective of industrial search or recommendation systems, a 0.2% AUC gain is not trivial, which can typically yield a 2% online CTR improvement (as in [4,33]).
>
> We declared in the introduction that GBA has comparable convergence properties with the synchronous mode (Line 58 of the old version). So we did not mean to claim that GBA has better model performance than sync mode. The main point/contribution of GBA is to design a novel method for switching between sync and async modes without sacrificing accuracy while not tuning hyper-parameters. The model performance of sync mode can be seen as the upper bound of GBA. Under the strained hardware resource, GBA can speed up 2.4x upon synchronous training while remaining the AUC scores. Meanwhile, GBA can provide a promising and stable accuracy compared to the other async baselines when switching between different training modes.
>
>
> **Question-1**:
> Figure 6 is hard to track, would it be possible to list the final AUC of each model for each task in a table?
>
> **Answer to Question-1**:
> We listed them in Appendix C and added more explanations in the updated version. Details can be found in the general response.

---

> > ### Comment · Reviewer_JyKb · 2022-08-08
> > **Thanks for your response!**
> >
> > Thanks for the author's detailed feedback!
> >
> > I appreciate the clarification of the experimental results and agree with the statements the author made. (I raised my score for this.)
> >
> > On the other hand, I am still a little confused about the claim of contribution in this paper. I would appreciate it if the author could correct me about my understanding of the contributions in this paper:
> >
> > - Discover a phenomenon that direct switch between synchronous and asynchronous training of recommendation model leads to diverge;
> >
> > - Some insight about why this phenomenon happens;
> >
> > - Based on these insights, design some algorithm to overcome this issues.
> >
> > Logically, how could one propose a solid solution without understanding the phenomenon?

---

> > > ### Author Response · Authors · 2022-08-09
> > > **Thanks again for your response and comments！**
> > >
> > > We have done many experiments and studies **to understand this phenomenon** in terms of different hyper-parameters. As we said in our last response, it is a common phenomenon encountered by our researchers / developers, which remains an open question in the community. And a comprehensive theoretical analysis of this phenomenon is **out of our scope**.  Following your suggestion, we have added a **theoretical analysis** of this phenomenon in terms of batch size in Appendix. However, our main contribution lies in the design of GBA to switch between training modes **without tuning hyper-parameters**. GBA **connects sync and async modes**, and it solves a big problem of our daily training under limited resources.

---

### Author Response · Authors · 2022-08-02
**General Response**

We appreciate all the reviewers for your valuable comments and suggestions. We have submitted an **updated version** of the paper based on these comments. We hope that the latest draft addresses your questions. The revisions are summarized as follows:

* We provide **theoretical analysis** in Appendix D to discuss the sudden drop of model accuracy after switching, which is based on our convergence analysis in Section 4.2. These analyses connect Observation 2 and Insight 1 in Section 3.2.
* We clarify the statements about Figure 6 in Section 5.2, and present the detailed statistics in Appendix C. Table R1-R3 depict the AUCs after inheriting the checkpoints trained by synchronous training. We collect the mean AUC for the first, last and all days of the three datasets, as shown in Table R4. We can infer from Table R4 that GBA provides the closest AUC score as synchronous training. GBA appears with the **lowest immediate AUC drop** after switching. Throughout the three datasets, GBA outperforms the other baselines by at least **0.2% (Hop-BW in Table R4)** when switching from synchronous training. More explanations can be found in the updated version.
* We polish the paper and fix some typos in the experiment section.

**Table R1: Figure 6(a) - Criteo (from Sync.)**

|Date|Sync.|GBA|Hop-BW|Hop-BS|BSP|Aysnc.|
|:-----|:-----|:-----|:-----|:-----|:-----|:-----|
|13|0.7999|0.7964|0.7954|0.7924|0.7930|0.5000|
|14|0.7957|0.7932|0.7959|0.7869|0.7886|0.5000|
|15|0.7967|0.7957|0.7895|0.7891|0.7889|0.5000|
|16|0.7963|0.7956|0.7932|0.5040|0.7891|0.5000|
|17|0.7962|0.7955|0.7930|0.5040|0.7883|0.5000|
|18|0.7957|0.7950|0.7939|0.5030|0.7862|0.5000|
|19|0.7972|0.7966|0.7968|0.5030|0.7863|0.5000|
|20|0.7974|0.7973|0.7985|0.5030|0.7868|0.5000|
|21|0.7965|0.7959|0.7948|0.5050|0.7863|0.5000|
|22|0.7957|0.7955|0.7939|0.5040|0.7865|0.5000|
|23|0.7987|0.7986|0.7933|0.5060|0.7871|0.5000|
|Avg.|0.7969|0.7959|0.7944|0.5819|0.7879|0.5000|

**Table R2: Figure 6(b) - Alimama (from Sync.)**

|Date|Sync.|GBA|Hop-BW|Hop-BS|BSP|Aysnc.|
|:-----|:-----|:-----|:-----|:-----|:-----|:-----|
|6|0.6490|0.6489|0.6472|0.6488|0.6472|0.5000|
|7|0.6503|0.6502|0.6478|0.6503|0.6500|0.5000|
|8|0.6523|0.6523|0.6483|0.6523|0.6512|0.5000|
|Avg.|0.6505|0.6505|0.6478|0.6505|0.6495|0.5000|

**Table R3: Figure 6(c) - Private (from Sync.)**

|Date|Sync.|GBA|Hop-BW|Hop-BS|BSP|Aysnc.|
|:-----|:-----|:-----|:-----|:-----|:-----|:-----|
|15|0.7877|0.7880|0.7870|0.7875|0.7880|0.7795|
|16|0.7874|0.7877|0.7860|0.7878|0.7870|0.7785|
|17|0.7856|0.7860|0.7840|0.7858|0.7850|0.7774|
|18|0.7884|0.7888|0.7850|0.7882|0.7877|0.7873|
|19|0.7894|0.7905|0.7855|0.7890|0.7886|0.7878|
|20|0.7785|0.7788|0.7750|0.7781|0.7774|0.7598|
|21|0.7865|0.7868|0.7823|0.7863|0.7850|0.7769|
|22|0.7862|0.7870|0.7825|0.7858|0.7862|0.7754|
|Avg.|0.7862|0.7867|0.7834|0.7861|0.7856|0.7778|

**Table R4: Average AUC decrement on three datasets between GBA and the other baselines (from Sync.).**

| |Sync.|Hop-BW|Hop-BS|BSP|Aysnc.|
|:-----|:-----|:-----|:-----|:-----|:-----|
|1st day|_+0.0011_|-0.0012|-0.0015|-0.0017|-0.1513|
|Last-day|_-0.0002_|-0.0046|-0.0979|-0.0045|-0.1542|
|Average|_+0.0002_|**-0.0025**|-0.0716|-0.0034|-0.1518|

---

### Meta-Review · Area_Chair_rJRC · 2022-08-23

**Recommendation:** Accept
**Confidence:** Certain

**Metareview:**

The paper identifies and illustrates a practically relevant challenge for training of deep learning-based recommender systems on distributed architectures: switching between synchronous and asynchronous training modes. The proposed mechanism called global batch gradient aggregation (GBA) is simple but mitigates the need to do hyper-parameter tuning when switching which was identified as the critical performance bottleneck. Experiments were conducted on industry-scale recommendation tasks and show that the proposed method is effective and improves in accuracy over fully asynchronous training methods and in speed over synchronous training schemes.

The prevailing opinion among the reviewers was that the paper is well written, well executed and addressed a practically relevant not previously studied problem. The identified challenge is clearly outlined and the proposed solution is well motivated, analyzed with ablation studies and shown to be effective. I advocate acceptance because it is a well executed practical piece of work.

For a potential camera ready version the authors should anticipate the questions of the reviewers and clarify them in the manuscript. They should also be more explicit about the manual steps and the parameter choices required when implementing the method so the limitations are more clear. Further, it would be appreciated if the authors would do an effort beyond acceptance of the paper to push the (currently proprietary) optimizer into an open source framework to make it available to the broader community as promised in the checklist.


**Award:**

No

---

### Decision · Program_Chairs · 2022-09-14

Accept